# The role of morphodynamics in predicting coastal flooding from storms on a dissipative beach with SLR conditions

Jairo E. Cueto[1,2], Luis J. Otero[3], Silvio R. Ospino-Ortiz[3,4], Alec Torres-Freyermuth[5]

[1] Coastal Geology and Sedimentology Group. Institute of Geosciences. Kiel University. Kiel, Germany.
[2] Research Group in Natural and Exact Sciences – GICNEX. Department of Natural and Exact Sciences. Universidad de la Costa. Barranquilla (Atlántico), Colombia.
[3] Research Group in Geosciences – GEO4. Department of Physics and Geosciences, Universidad del Norte. Barranquilla (Atlántico), Colombia.
[4] Centro de Investigación de Ingeniería de Cormagdalena – CIIC. Barranquilla (Atlántico), Colombia.
[5] Coastal Processes and Engineering Laboratory– LIPC, Engineering Institute. Universidad Nacional Autónoma de México (UNAM). Sisal (Yucatán), México.

*Correspondence to*: J. Cueto (jcueto7@cuc.edu.co)

**Abstract.** We investigate the role of morphodynamic changes in the flooding of a micro-tidal dissipative beach for both current and sea level rise scenarios. By considering beach morphodyanmics and flood processes associated with highly energetic waves, the study allows to evaluate threats to coastal zones. Coupling of SWAN and XBeach models are employed to propagate offshore wave conditions to the swash zone, estimating morphological changes and flooding associated to wave conditions during cold fronts and hurricanes that affected Cartagena de Indias (Colombia). The numerical models were calibrated from previous research in the study area. The results indicate that numerical modelling of flooding on microtidal dissipative beaches under extreme wave conditions should be approached by considering beach morphodynamics, because ignoring them can underestimate flooding by ~15%. Moreover, model results suggest that beach erosion and flooding are intensified by sea level rise, resulting in the most unfavourable condition when extreme events are contemporaneous with high tides. In this case, the increase in erosion and flooding is ~69% and ~65%, respectively, when compared with the present conditions of sea level.

## 1 Introduction

The impact of extreme storms on a coast has adverse consequences for coastal communities associated with loss of life and infrastructure damage, as well as significant indirect economic losses (Kron, 2013; Bertin et al., 2014; Sills et al., 2008). In highly urbanized coastal areas, such as Cartagena de Indias (Colombia), where residence and industries are located near the coast, such storms generally damage or destroy the infrastructure. These effects are the integrated consequences of two storm-induced coastal hazards, flooding and erosion (Sallenger et al., 2000; Sanuy and Jiménez, 2019; Guimaraes et al., 2015). In this context, an adequate quantification of these hazards is an essential part of risk management (e.g., Ciavola et al., 2011; Jiménez et al., 2018; Plomaritis et al., 2018; Harley et al., 2017; Sanuy and Jiménez, 2019).

The use of process-oriented numerical models to forecast storm-induced morphodynamic changes in given scenarios is a widespread and widely accepted methodological practice (e.g., Roelvink et al., 2009; McCall et al., 2010; Dissanayake et al.,

2014; Annette et al., 2020). Predicting flooding associated with storm impacts accompanied by intense erosion can pose a problem on multiple scales, governed by complex interactions between a great variety of hydrodynamic processes and of sediment transport (Wu et al., 2011; Christensen et al., 2013; He et al., 2015). Among the most important factors for the prediction are: (i) increase in sea level associated with climate change, which increases the exposure of the coast during extreme events; (ii) increase in intensity and frequency of storms events (IPCC, 2021); (iii) beach erosion; (iv) flooding during and

after a storm (Elsayed and Oumeraci, 2016).

The traditional approach for the study of flooding and erosion induced by storms has been to investigate the two coastal threats separately, i.e., flooding (e.g., Ruju et al., 2014; Guimaraes et al., 2015; Medellín et al., 2016; Lerma et al. 2017; Fiedler et al., 2018) and erosion (Schambach et al., 2018). Particularly for the area of the Colombian Caribbean coast, works have focused

on evaluating the threat of flooding from extreme waves, without considering changes in beach morphology produced by the studied storm (Andrade et al., 2013; Orejarena et al., 2019). However, recent studies have examined the impact of both threats simultaneously using the eXtreme Beach Behavior (XBeach) model as a tool (Stockdon et al., 2014; de Santiago et al., 2017; Sanuy and Jiménez, 2019; Enríquez et al., 2019).

Extreme waves on the Colombian Caribbean coast are associated with hurricanes and cold fronts (Ortiz-Royero, 2007; Ortiz-Royero et al., 2013; Otero et al., 2016; Cueto and Otero, 2020). The meteo-marine event that more severely affected the central coast of Colombia in the last decade was the cold front of March 2009, which collapsed a 200-m section of the old Puerto Colombia pier (Ortiz-Royero et al., 2014). Although Colombia is in a fortunate location in the Caribbean regarding the trajectory of hurricanes, it is not exempt from the consequences of a devastating passage. Hurricanes such as Joan (1988)

(Ortiz-Royero, 2008), Lenny (1999) (Ortiz-Royero, 2009), and more recently Matthew (2016) (Cueto and Otero, 2020) and Iota (2020), among others, have impacted the Colombian Caribbean coasts, causing flooding and erosion along the coastline. After an analysis of 1980–2014 satellite images and field measurements, Rangel-Buitrago et al. (2015) estimated that 48.3% (1182 km) of the Caribbean Colombian coast had serious erosion problems. Only 33.2% (812.6 km) could be considered stable and the remaining 18.4% (450.5 km) showed accretion. The erosive dynamics found along the northern coasts of Colombia

have been influenced by different factors, among which are storms (Rangel-Buitrago et al., 2015; Otero et al., 2016). In the particular case of the coastal area of Cartagena de Indias, the coastline experienced setbacks of as much as 50 m because of an anomalous succession of storms (hurricanes and cold fronts) occurring between January 2010 and January 2011 (Bernal et al., 2016). This forced the implementation of an emergency procedure for recovery of the beaches using hydraulic fill. Moreover, floods caused by dry-season cold fronts affecting the city of Cartagena de Indias have become increasingly frequent (Andrade

et al., 2013; Otero et al., 2016). This is because it has been established that the rate of rise in mean sea level in this region was 5.6 mm/year during the period 1950–2009 (Torres and Tsimplis, 2012), which has aggravated the impact of the aforementioned phenomena in the region (Orejarena et al., 2019).

Per the above, the aim of the present work was to study the role of morphodynamics in the flooding of a dissipative beach with microtidal regime. Moreover, the simultaneous and individual effects of erosion and flooding in scenarios of long-term mean sea level rise were considered. For this analysis, we selected a sector of the Colombian Caribbean coast with great touristic, historical, economic, cultural and social importance, namely, Cartagena de Indias. By simultaneously considering erosion and flood processes associated with highly energetic waves, the numerical model allows us assessing threats to coastal zones and to investigate the implications of not including the effects of morphodynamic changes on the flooding. This will enable the design of early warning systems, in order to protect the population and infrastructure from threats by the sea.

## 2 Description of study area

The study area is located in Bocagrande beach (Figure 1), which forms a section of the coastline of the city of Cartagena de Indias in the central Colombian Caribbean. Regarding morphology, Bocagrande has a dissipative profile (Figure 2), with an offshore slope of 0.3% and foreshore slope of 1.8%. Furthermore, it is composed of fine sands with grain sizes between 0.08 and 0.42 mm (Conde *et al.,* 2017a; b). Within the study domain of Bocagrande, there are six groins perpendicular to the coast that extend between 50 and 100 m within the breaker zone . The tide in the study area fluctuates between 20 and 30 cm, classifying Bocagrande as a microtidal beach (Restrepo *et al.,* 2012; Restrepo *et al.,* 2016). Atmospheric-induced surges can exceed 20 cm when highly energetic storms occur in the area (Andrade et al., 2013).

Owing to its geographical location in the Caribbean, the hydro-climatology of the study area is influenced by the migration of the Intertropical Convergence Zone (ITCZ) (Poveda, 2004; Pérez *et al.,* 2018). The ITCZ is an area of the globe where the trade winds of the northern and southern hemispheres converge, generating a low-pressure belt around the equator that oscillates seasonally. The Colombian Caribbean is governed climatically by this movement (Poveda, 2004), producing a bimodal regime with two dry seasons (December–March and June–July) and two wet seasons (April–May and August–November). During dry periods, winds tend to be stronger; from December to March, the trade winds from the north predominate, while between June and July the prevailing winds are from the southeast. In contrast, the rainy months have weaker winds. The influence of the trade winds in times of drought produces waves with strong energy, the northeast direction being the component with the highest probability of occurrence (32%) (Ortiz-Royero, 2012; Restrepo *et al.,* 2012). With less probability (~15%), waves arrive from the east-northeast, north and north-northeast directions. Waves originating outside these directions are generated by local winds and have a low probability of occurrence in the area.

Detailed bathymetries were measured along with control profiles in field campaigns performed before and after a succession of cold fronts between November 2014 and February 2015. These bathymetries were constructed using data from an ODOM Hydrotrac 2 single beam echosounder (submerged areas) and a high-resolution LIDAR topography (dry beach and urban areas). All vertical heights in the Digital Terrain Model (DTM) were referenced to average low tides of syzygy (MLWS) and

horizontal coordinates were referenced to UTM 18N, following the standards of the International Hydrographic Organization (IHO). Hydrodynamic data were also acquired during field campaigns (four days of measurements each campaign), using a cross-shore arrangement of four Aquadopp current meters (S5, S4, S3 and S2) and an RBR pressure sensor (S1). The averaged significant Sea-Swell wave heights during the wet season field campaign at S1, S2, S3, S4 and S5 were 1.2, 0.4, 0.4, 0.3 and

0.2 m, respectively; for the dry season campaign these values were 1.7, 0.7, 0.6, 0.5 and 0.4 m, respectively. Peak periods averaged 8 s (wet season campaign) and 9 s (dry season campaign) at the outermost sensor (S1). The predominant incident wave direction was north-northeast for both measuring periods. A detailed description of the sections referring to these field campaigns is given in Cueto and Otero (2020).

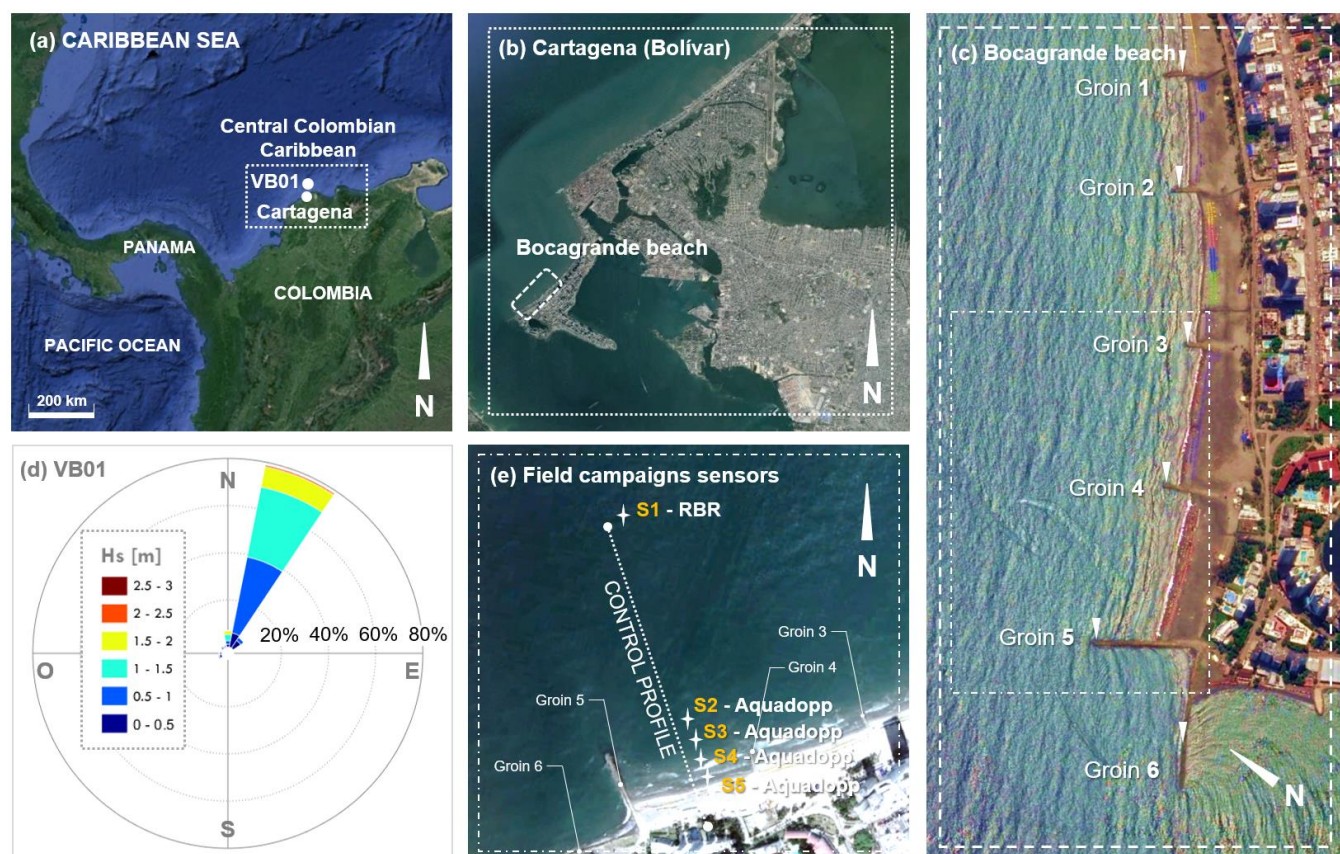

**Figure 1: Location of study area: (a) location of Cartagena in the Colombian Central Caribbean and Virtual Buoy 01 [VB01] (10°40′1.12″N, 75°30′0.00″W); (b) location of Bocagrande Beach in Cartagena de Indias; (c) studied area (computational domain of XBeach) of Bocagrande beach; (d) wave rose extracted from VB01; (e) instrumental setup during field campaigns (S1: Sensor 1 – RBR; S2, S3, S4 and S5: Sensors 2, 3, 4 and 5, respectively – Aquadopps) – the reader is referred to Cueto and Otero (2020) for a detailed description of sensors location, depth and measurement rates. Map base images retrieved**

**from Google Earth © and modified by the authors.**

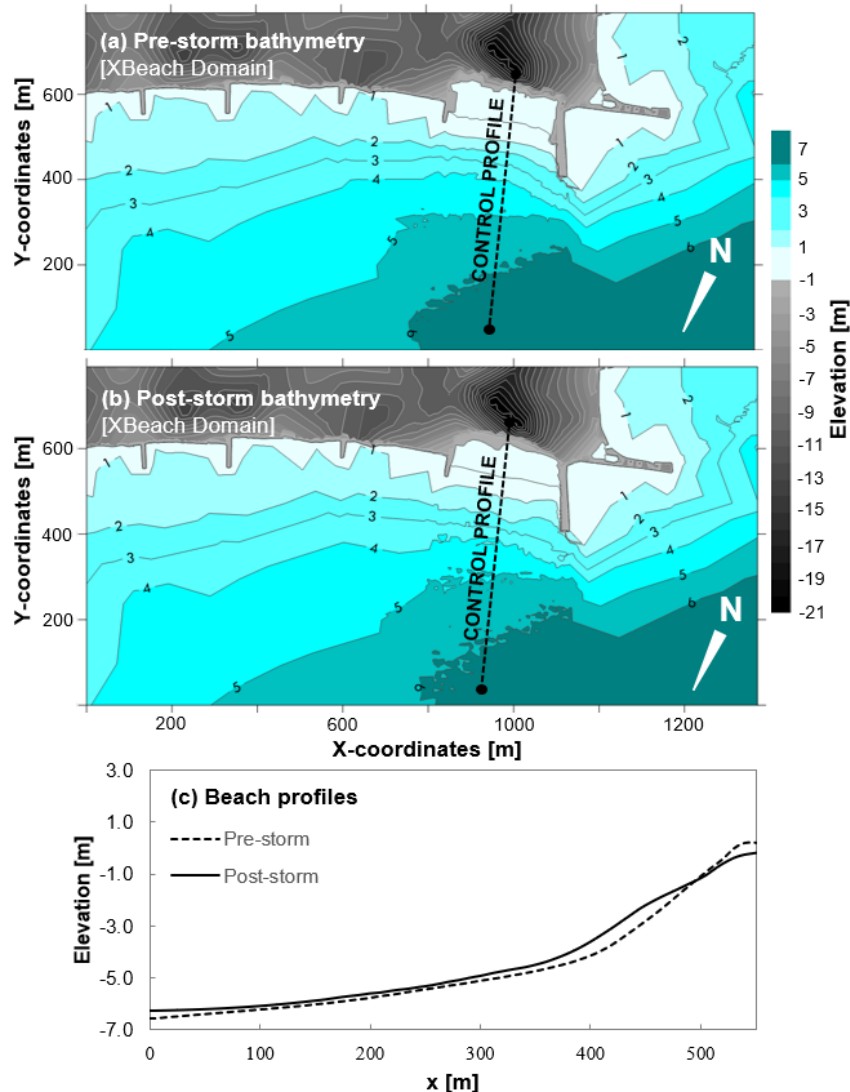

**Figure 2: (a) Pre-storm detailed bathymetry; (b) post-storm detailed bathymetry; (c) pre- and post-storm beach profiles at Bocagrande (Cartagena de Indias).**

## 3 Methodology

### 3.1 Selected events

In order to identify extreme wave events in Bocagrande, wave series were extracted from the Virtual Buoy 01 (VB01 – located at 800 m depth) in the central Colombian Caribbean, near the coast of Cartagena de Indias (Figure 1a). These wave reanalysis time series were obtained from a NOAA database that used the third-generation model WAVEWATCH III (Chawla et al., 2013). Vega (2017) these wave series to the conditions of the Colombian Caribbean by using a hybrid methodology of wave

reanalysis scale refinement. This approach included the effect of specific spectral wave, wind and bottom roughness parameters for the studied area after conducting a sensitivity analysis. It is well known that the wave reanalysis information tends to underestimate wave heights associated with extreme events within the Caribbean (Ortiz-Royero, 2009). However, with the adjustment of Vega (2017), errors do not exceed 5% for significant wave height calculations.

Subsequently, the series extracted for each area of interest were compared with the exact dates on which hurricanes and cold fronts occurred in the Colombian Caribbean over the last four decades (Ortiz-Royero, 2012; Ortiz-Royero et al., 2013; Bernal et al., 2016; Otero et al., 2016). We thereby constructed a detailed inventory of extreme wave events in the study area. From this information, we selected case studies for subsequent hydro-morphodynamic modelling. As selection criteria, we considered events that strongly affected the Cartagena de Indias area, analyzing the energy contents, impacts, and proximity

to the coast of each storm (Ortiz-Royero, 2012; Ortiz-Royero et al., 2013; Bernal et al., 2016; Otero et al., 2016). To investigate the combined effects of erosion and flooding following the impact of storms in future scenarios with higher sea levels, the sea level rise (SLR) conditions for the coming decades at Bocagrande beach predicted and discussed by Orejarena et al. (2019) will be considered for the selected events.

The inventory with some of the most important hurricanes and cold fronts affecting Cartagena over the last 40 years is shown in Table 1. Among the data from VB01, there are noteworthy maximum wave heights of 2.76 and 3.53 m. These values were recorded during the passage of Hurricane Lenny (1999) and the cold front of December 2017, respectively. Lenny traversed the study area as a tropical storm, producing extreme waves during approximately two days. In contrast, the cold front of December 2017 persisted longer in the study area, generating an increase in wave heights over five days. Analogously, the

extreme events of 2010 reported by Bernal et al. (2016) were chosen for case study. According to Bocagrande's historical records, this set of storms generated unprecedented recessions along the coastline (50 m in one year) and was particularly associated with the 2010 Atlantic hurricane season (Bernal et al., 2016). We especially emphasize three cold fronts during March 2010 (now on referred as cold fronts X, Y and Y), which most influenced the erosive processes of that year (25–30 m retreat in less than a month). Between the first (X) and the second cold front (Y) of that month there was a period of six days

with moderate waves. Between the second (Y) and third cold front (Z) there were moderate waves over eight days. SLR was considered for future scenarios following the projections of Orejarena et al. (2019) for the Cartagena de Indias area. These projections were prepared according to data from the Integrated Climate Data Center of the University of Hamburg. The aforementioned authors projected a SLR of 0.11 (2025), 0.24 (2050), 0.38 (2075), and 0.52 m (2100) at Bocagrande. The selected events from which the case studies were formulated are shown in italics in Table 1.


**Table 1: Analyzed storms from VB01 (Bocagrande) wave series. Duration over the studied area, maximum significant wave height (H$_s$), peak period (T$_P$) and mean direction (D$_m$) are displayed. Selected events are shown in bold letters.**

| Hurricane | Duration (days) | VB01 | | |
|---|---|---|---|---|
| | | H$_s$(m) | T$_p$(s) | D$_m$(°) |
| Joan (1988) | 2 | 1.31 | 4.98 | 259.92 |
| Bret (1993) | 2 | 1.90 | 8.77 | 17.81 |
| Cesar (1996) | 2 | 1.48 | 9.38 | 17.69 |
| Mitch (1998) | 2 | 1.35 | 6.62 | 258.61 |
| **Lenny (1999)** | **2** | **2.76** | **7.91** | **316.09** |
| Sandy (2012) | 2 | 1.70 | 6.63 | 260.37 |
| Matthew (2016) | 3 | 2.29 | 12.03 | 192.07 |
| Cold fronts | | H$_s$(m) | T$_p$(s) | D$_m$(°) |
| Cold front (March, 2009) | 5 | 2.49 | 8.96 | 26.76 |
| **Cold front X (March, 2010)** | **5** | **2.66** | **7.51** | **357.24** |
| **Cold front Y (March, 2010)** | **3** | **1.34** | **5.90** | **24.76** |
| **Cold front Z (March, 2010)** | **6** | **1.60** | **8.81** | **25.87** |
| **Cold front (December, 2017)** | **5** | **3.53** | **9.14** | **35.79** |


## 3.2 Numerical modelling

This section outlines the numerical modelling approach adopted here. To quantify the storms impact, we employed the Simulating WAves Nearshore (SWAN) numerical model (Boij and Holthuijsen, 1999) for the wave propagation from deep waters and its subsequent nesting with the XBeach model (Roelvink et al., 2009) to simulate beach hydrodynamics and

morphodynamics in Cartagena de Indias.

### 3.2.1 *From deep waters to the nearshore: Wave propagation using SWAN*

Propagation from deep waters during the selected events was determined by the SWAN model, originally developed by Booij and Holthuijsen (1999). The wave parmaeters time series corresponding to the selected events were propagated from the location of virtual buoy VB01(800 m depth) in deep waters (Figure 1a) to ~1 km from the Bocagrande coast (which is the

offshore boundary of the computational domain used in XBeach, which is between 5.5 – 6.5 m depth).

SWAN model was calibrated for the study area by Cueto and Otero (2020), considering in situ wave data measured in field campaigns before and after storms, following parameters established by Conde et al. (2017a). Table 2 shows the computational domain used for SWAN modelling of Cartagena, and Table 3 the error parameters calculated by Cueto and Otero (2020) for

model performance at Bocagrande.

**Table 2: SWAN and XBeach computational domain for Bocagrande Beach.**

| GRID | SWAN | XBeach |
|---|---|---|
| Cell size (cross-shore x alongshore) [m] | 100x100 | 5x5-3x5-1x5 |
| X [nodes] | 334 | 790 |
| Y [nodes] | 280 | 1570 |
| Area (km²) | 935.2 | 1.24 |
| Offshore max. depth [m] | 830 | 6.5 |

**Table 3: SWAN error parameters for Bocagrande (Cueto and Otero, 2020).**

| Parameter | Bocagrande | |
|---|---|---|
| | Pre-storm | Post-storm |
| Bias | −0.05 | 0.01 |
| $r^2$ | 0.91 | 0.87 |
| Willmott | 0.96 | 0.95 |

### 3.2.2 Beach morphodynamics and coastal flooding: XBeach

Beach hydrodynamics and morphodynamics and coastal flooding during extreme wave events, considering SLR scenarios at Bocagrande, were estimated using XBeach (Roelvink et al., 2009).

XBeach is an open-source numerical model originally developed to simulate hydrodynamic and morphodynamic processes on sandy beaches, using a domain of kilometers and the timescale of storms. This includes hydrodynamic processes such as (Sea – Swell) SS wave transformation (refraction, shoaling and breaking), infragravity wave (generation, propagation, and dissipation), wave-induced setup and non-stationary currents, and overwash and flooding. The morphodynamic processes that XBeach solves include suspended sediment and bottom transport, dune erosion, bottom updating, and breaching (Roelvink et al., 2009). The model can be used in three modes, i.e., phase-averaged, surfbeat and non-hydrostatic. In the present study, we used the surfbeat mode for representing the conditions of hurricanes and cold fronts on Bocagrande. The surfbeat mode is mostly used to account the impact of storms with highly energetic conditions on dissipative beaches, where the Sea-Swell waves are largely dissipated by the time they are close to the coastline. In this mode, XBeach calculates the morphological processes and run-up with the infragravity wave band accounting indirectly for the contribution of the short waves. It does not solve Sea-Swell waves individually (as in non-hydrostatic mode). The model includes solvers for non-linear shallow water equations (NLSWE) and morphodynamics, so its application can be extended to simulate coastal flooding together with erosion

processes, using a single mesh calculation for both modules. This approach offers the advantages of simulating the mutual interaction between hydrodynamics and morphodynamics.


The computational domain defined for hydro-morphodynamic modelling of Bocagrande Beach using XBeach (Figure 2) covers an area of 1.24 km$^2$ (1.57 km alongshore and 0.79 km cross-shore). The domain is distributed on a mesh of rectangular cells with variable sizes, ranging from 5 m wide in the areas furthest from the coast to 1 m in the closest areas. Characteristics of the computational domain used with XBeach are shown in Table 2.


The model was previously calibrated based on experimental data (topo-bathymetric and hydrodynamic) collected during pre- and post-storm field campaigns. Cueto and Otero (2020) described this calibration for the study area in detail, following the two-step methodology presented by Nederhoff (2014) and aspects from more extensive calibrations carried out by Ranasinghe et al. (2011) and Luijdendijk et al. (2017) (Table 4). The latter calibration procedures also include morphological processes on

timescales greater than those of storms. Table 4 shows a summary of the calibration procedure results. The best morphologic representations were obtained when using a *facua* (factor $u_a$ – related to the advection velocity of the sediment) parameter of 0.35 and 0.45, combined with a Chezy friction coefficient of 45 m$^{1/2}$/s. This Chezy friction value is directly related to the sediments' characteristics of the studied beach, which is mostly constituted by fine sands with grain sizes within the range of 0.08 and 0.42 mm (Conde et al., 2017a; b). According to the tests conducted by Cueto and Otero (2020), an increase (decrease)

in the bottom friction through the Chezy coefficient would cause greater (lower) dissipation of the incident waves energy, leading to an underestimation (overestimation) of on Bocagrande's morphology fluctuations. According to the classification elaborated by van Rijn (2003), which includes a qualitive scale for morphological representation based on different Brier Skill Scores (BSS) (BSS = 1, perfect representation; BSS = 0, poor representation), the model approximation was qualified as "Excellent".


**Table 4: XBeach morphodynamic parameters and performance for Bocagrande, Cueto and Otero (2020).**

| Test | *facua* | Chezy [m$^{1/2}$/s] | BSS | Qualification |
|------|---------|---------------------|------|---------------|
| 1 | 0.05 | 45 | 0.41 | Reasonable |
| 2 | 0.20 | 45 | 0.67 | Good |
| 3 | 0.35 | 45 | 0.89 | Excellent |
| 4 | 0.45 | 45 | 0.80 | Excellent |

For the hydro-morphodynamic modelling with XBeach, the case studies shown in Table 5 were established. Case studies are derived from the selected events: Lenny 1999 ("A" cases), the cold fronts of 2010 ("B" cases – that include the cold fronts X,

Y and Z) and the cold front of 2017 ("C" cases). Present conditions of sea level (cases A1, B1 and C1) and the future projections of SLR for 2025 (+0.11 m – cases A2, B2 and C2) and 2050 (+0.24 m – cases A3, B3 and C3) addressed by Orejarena et al.

(2019), were included. Scenarios A4, B4 and C4 were set to analyse the effect of high tides on erosion and flooding processes combined with SLR. The morphological updating in XBeach was turned on and off for each case study. In this way, the influence of erosive processes on flooding was checked. Flooding extent in all scenarios is estimated from the run-up output
that XBeach calculates directly. The maximum reach of flooding in urban areas is also accounted by calculating the water intrusion after the first street parallel to Bocagrande beach (exposed as a thin red line in Figures 3 – 6). The models included a non-erodible layer to simulate the hard structures present on Bocagrande beach. The time between the cold fronts X, Y and Z of 2010 was simulated using the stationary mode of XBeach as wave conditions during those intervals were low-energetic (~0.7 – 0.8 m). The sea levels of 2025 and 2050 for Cartagena were allocated at the offshore boundary as water level forcing.
The SLR was added to the water level input for the different case studies conditions. This approach of water levels was also used when setting up the SWAN model for the studied area.

**Table 5: Evaluation of flooding through numerical modelling of Bocagrande Beach. * The additional effect of a +0.25 m high tide was included, without considering the storm surge. ** Between cold fronts X and Y there was a six-day period of average wave**
**conditions, and eight days between Y and Z.**

| Case study | Storm | SLR (m) | $H_s$ (m) | $T_p$ (s) | Dm (°) | Duration (days) |
|---|---|---|---|---|---|---|
| A1 | | 0.00 (present conditions) | | | | |
| A2 | Lenny 1999 | 0.11 (2025) | 2.8 | 7.9 | 316 | 2 |
| A3 | | 0.24 (2050) | | | | |
| A4 | | 0.24* (2050) | | | | |
| B1 | | 0.00 (present conditions) | | | | |
| B2 | | 0.11 (2025) | | | | |
| B3 | Cold fronts (X+Y+Z) 2010 | 0.24 (2050) | (X) 2.7 (Y) 1.3 (Z) 1.6 | 7.5 5.9 8.8 | 357 24 25 | 5(X)+3(Y)+6(Z)** |
| B4 | | 0.24* (2050) | | | | |
| C1 | | 0.00 (present conditions) | | | | |
| C2 | Cold front 2017 | 0.11 (2025) | 3.5 | 9.1 | 35 | 5 |
| C3 | | 0.24 (2050) | | | | |
| C4 | | 0.24* (2050) | | | | |

## 4 Results

Figures 3–6 show the maximum extent of flooding (with and without morphodynamics) and post-storm shorelines simulated
by XBeach corresponding to the passage of storms with characteristics similar to Lenny 1999, the 2010 cold front sequence
and 2017 cold front, incorporating different SLR scenarios and using the current Bocagrande bathymetry. Figure 7 shows the
control profile of Bocagrande Beach evaluated for different cases resulting on beach retreats. Table 6 summarizes the most
important results of each case study.

### 4.1 Hurricane Lenny 1999

In case study A1 (Figure 3a), the coastline retreat varied between 15 and 18 m, with erosion more evident along beach sections
between groins 3 and 4 and 4 and 5. In case studies A2 (Figure 3b) and A3 (Figure 3c), with SLRs of +0.11 and +0.24 m
respectively, the erosive processes became larger, especially in the latter case. The coastline retreated between 19 and 24 m in
the XBeach simulations for case A2, whereas in the most extreme scenario (A3), erosion was able to cause the loss of as much
as 32 m of beach in the most critical section (between groins 4 and 5).

The maximum flooding extent in all case studies was greater for simulations that included morphologicall changes. When the
morphological updating was turned on the maximum range of the water layer during the storm was 95, 110 and 135 m
(measured from the shoreline) for cases A1, A2 and A3, respectively. In cases A2 and A3 (both with morphodynamics) the
results showed partial flooding in the urban area of Bocagrande, penetrating 31 m with the SLR conditions of 2025, and 41 m
with the SLR of 2050 (measured from the first street located next to the beach). Flooding in simulations without
morphodynamics did not extend beyond that boundary street.

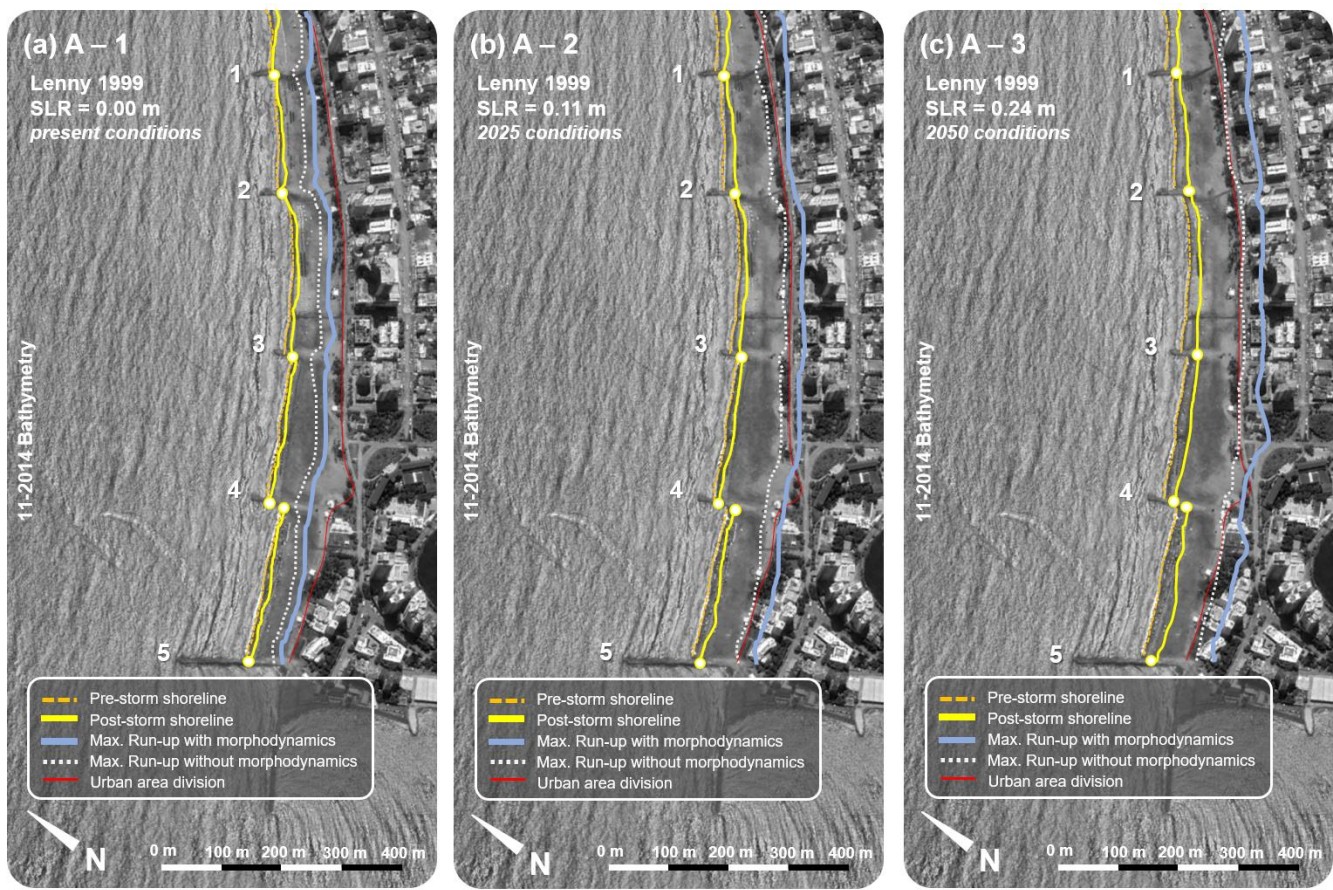

**Figure 3: Maximum flood range (with and without morphodynamics) and post-storm shorelines simulated by XBeach for a storm similar to Lenny 1999 with different SLR conditions. Map base images retrieved from Google Earth © and modified by the authors.**

### 4.2 Cold fronts sequence (2010)

The results of case B1 (Figure 4a) indicate maximum retreats of nearly 30 m in the section between groins 4 and 5. In the remaining Bocagrande sections, erosion consumed an average 25 m of beach. Upon applying the 2025 SLR conditions in the model (Figure 4b), the losses of coastline began to exceed 35 m and the erosion was accentuated in every Bocagrande section. In the most severe case (B3) (Figure 4c), the retreats reached ~50 m, as occurred between groins 2 and 3 in the most critical transect.

Regarding maximum flood extent, XBeach calculated that with an SLR of +0.24 m and a succession of cold fronts similar to that of 2010 (B3), the water line penetrated widely in the urban area of Bocagrande (96 m measured from the beginning of the first street with morphodynamics, and 67 m without morphodynamics). This case study had the greatest erosion and flooding in the analysis area. Measured from the shoreline, the maximum range of the water sheet with the morphological updating

activated was 109, 141 and 184 m in cases B1, B2 and B3, respectively. In cases B1 and B2, flooding also occurred in the urban area when the morphological updating was used, with maximum intrusions up to 11 m in the first case and 50 m in the second (measured from the first street). Without morphodynamics, the flood penetrated 94 m inland in case B1 (not exceeding the urban limit) and 113 m in B2 (both measured from the coastline).

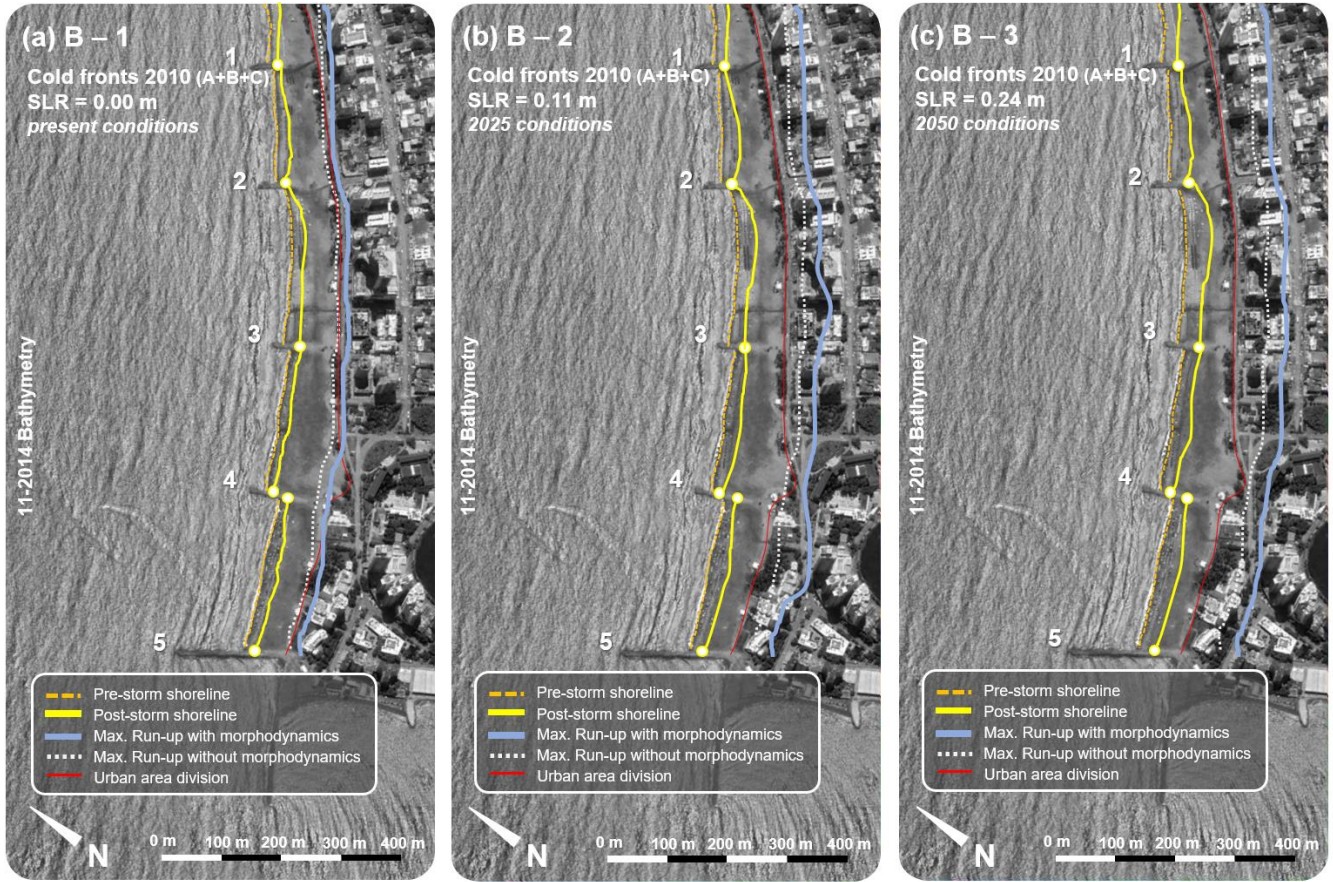

**Figure 4: Maximum flood range (with and without morphodynamics) and post-storm shorelines simulated by XBeach for 2010 cold front succession with different SLR conditions. Map base images retrieved from Google Earth © and modified by the authors.**

## 4.3 2017 cold front

From analysis of a storm similar to the cold front of 2017 with varying SLR conditions, the greatest coastline retreat calculated by XBeach exceeded 30 m. This was in the section between groins 2 and 3, after applying a SLR of +0.24 m (case C3, Figure 5c). In cases C1 (Figure 5a) and C2 (Figure 5b), beach losses calculated by the model were smaller, 19 and 26 m, respectively, in the sections where erosion was more severe.

The flooding under current SLR conditions reached 97 m and 88 m with and without morphodynamics, respectively, failing to reach the streets of Bocagrande. On the other hand, the maximum range of the water layer was 113 m (94 m without morphodynamics) under 2025 SLR conditions, and 148 m (119 m without morphodynamics) under SLR 2050 conditions. In cases C2 and C3, with morphodynamics, the water penetrated 36 and 49 m beyond the beginning of the first street.

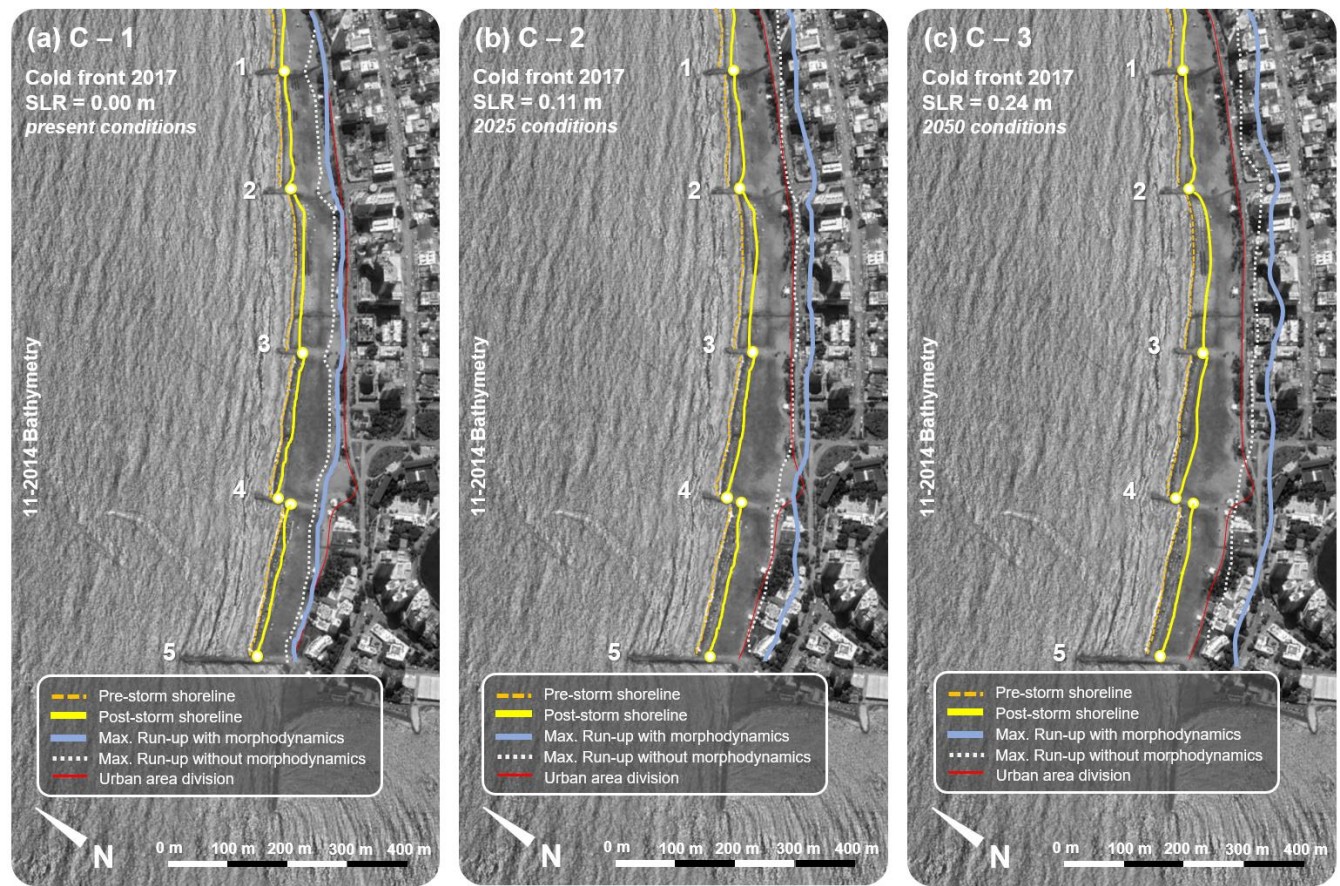

295

**Figure 5: Maximum flood range (with and without morphodynamics) and post-storm shorelines simulated by XBeach for a storm similar to 2017 cold front with different SLR conditions. Map base images retrieved from Google Earth © and modified by the authors.**

### 4.4 Storms with high tide +0.25 m

300    In case study A4 (Figure 6a), which corresponds to Hurricane Lenny with a SLR of +0.24 m plus high-tide effects (+0.25 m), the coastline receded between 20 and 34 m, erosion becoming more aggressive in beach sections between groins 3 and 4 and 4 and 5. In the case of the 2010 succession of cold fronts, including a SLR projected for the year 2050 and high tide (B4) (Figure 6b), erosion was severe, with beach losses up to and exceeding 50 m in most sections. The results for the 2017 cold front with SLR of +0.24 m and high tide (Figure 6c) indicate notable retreats of as much as 40 m in the beach sections between

305    groins 2 and 3, 3 and 4, and 4 and 5.

The results also show that the maximum flood extent occurred when the morphological updating was activated in XBeach. The sea produced inland flooding as far as 147, 199 and 157 m in cases A4, B4 and C4, respectively, with morphodynamics. Likewise, for all cases, in this section there was penetration of the sea into urban areas of Cartagena, reaching 52 (A4), 105 (B4) and 58 m (C4) as measured from the boundary street of the area.

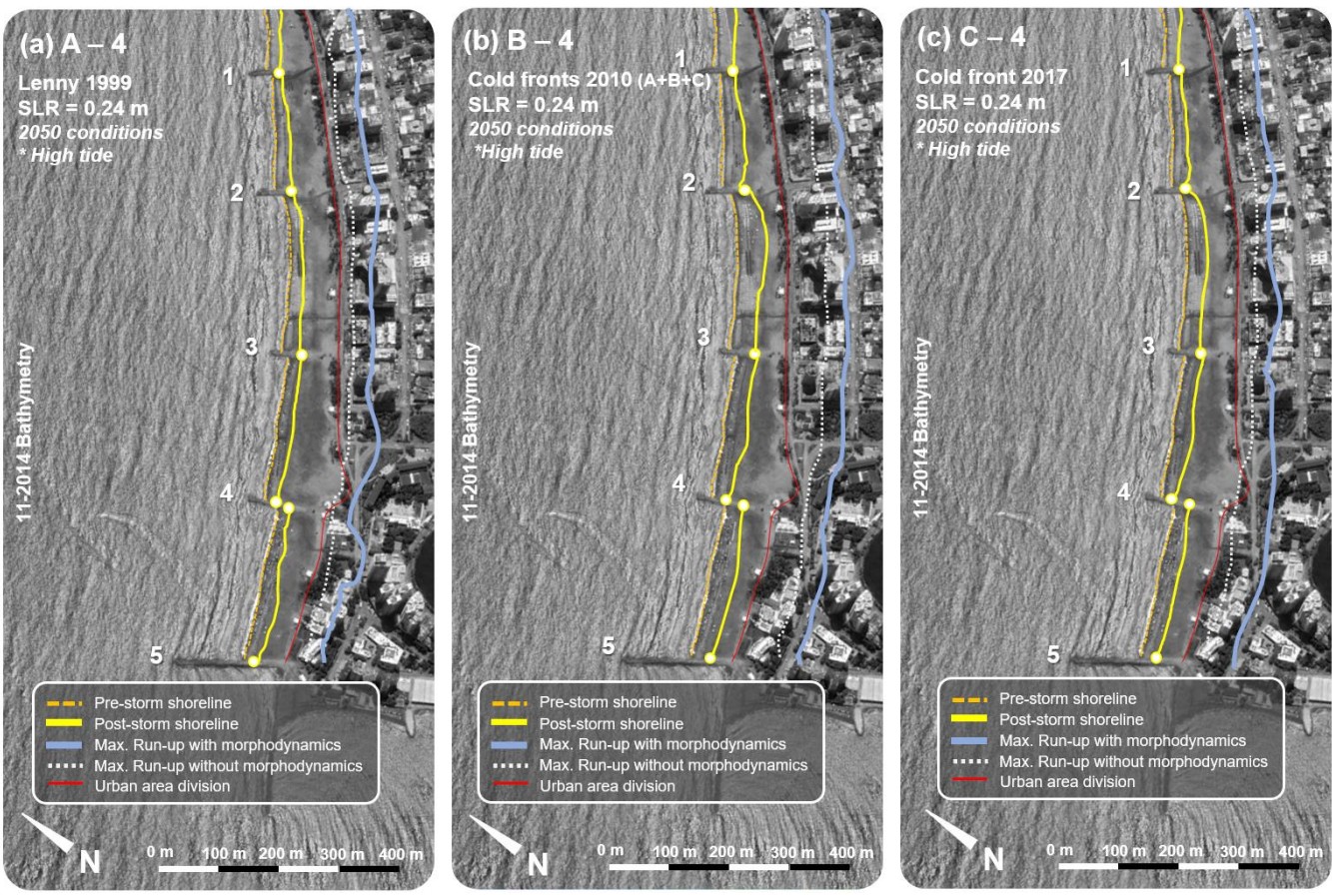

**Figure 6: Maximum flood range (with and without morphodynamics) and post-storm shorelines simulated by XBeach including additional effect of a +0.25 m high tide for Lenny 1999 (a), 2010 cold front succession (b), and 2017 cold front (c) with different SLR conditions. Map base images retrieved from Google Earth © and modified by the authors.**

**Table 6: Maximum coastline retreats, maximum flood reach (with and without morphodynamics) and maximum penetration into urban area calculated by XBeach for the case studies. *Indicates penetration into urban area considering morphodynamics only.**

| Case study | Max. beach retreat [m] | Max. inundation reach (*with morph.*) [m] | Max. inundation reach (*without morph.*) [m] | Max. penetration in urban area* [m] |
|:---:|:---:|:---:|:---:|:---:|
| A1 | 18 | 95 | 82 | 0 |
| A2 | 24 | 110 | 91 | 31 |
| A3 | 32 | 135 | 116 | 41 |
| A4 | 34 | 147 | 127 | 52 |
| B1 | 29 | 109 | 94 | 11 |
| B2 | 36 | 145 | 113 | 50 |
| B3 | 49 | 184 | 151 | 96 |
| B4 | 52 | 199 | 164 | 105 |
| C1 | 24 | 97 | 90 | 0 |
| C2 | 29 | 113 | 94 | 36 |
| C3 | 35 | 148 | 119 | 49 |
| C4 | 40 | 157 | 131 | 58 |

## 4.5 Post-storm beach profile changes

Figure 7 shows the evolution of Bocagrande's control profile for the selected case studies using XBeach. Coastal line retreats were 15 (A1), 22 (A2) and 30 m (A3) when Lenny 1999 was simulated (Figure 7a). For the successive cold fronts of 2010, the model calculated beach losses of 28 (B1), 33 (B2) and 45 m (B3) (Figure 7b). Retreats of 23 (C1), 28 (C2) and 34 m (C3) were calculated after the simulation of the cold front of 2017 (Figure 7c). Finally, XBeach calculated coastal line retreats of 31 (A4), 40 (B4) and 49 m (C4) for Lenny 1999, the cold fronts of 2010 and 2017, respectively, when the high-tide effects were included (Figure 7d).

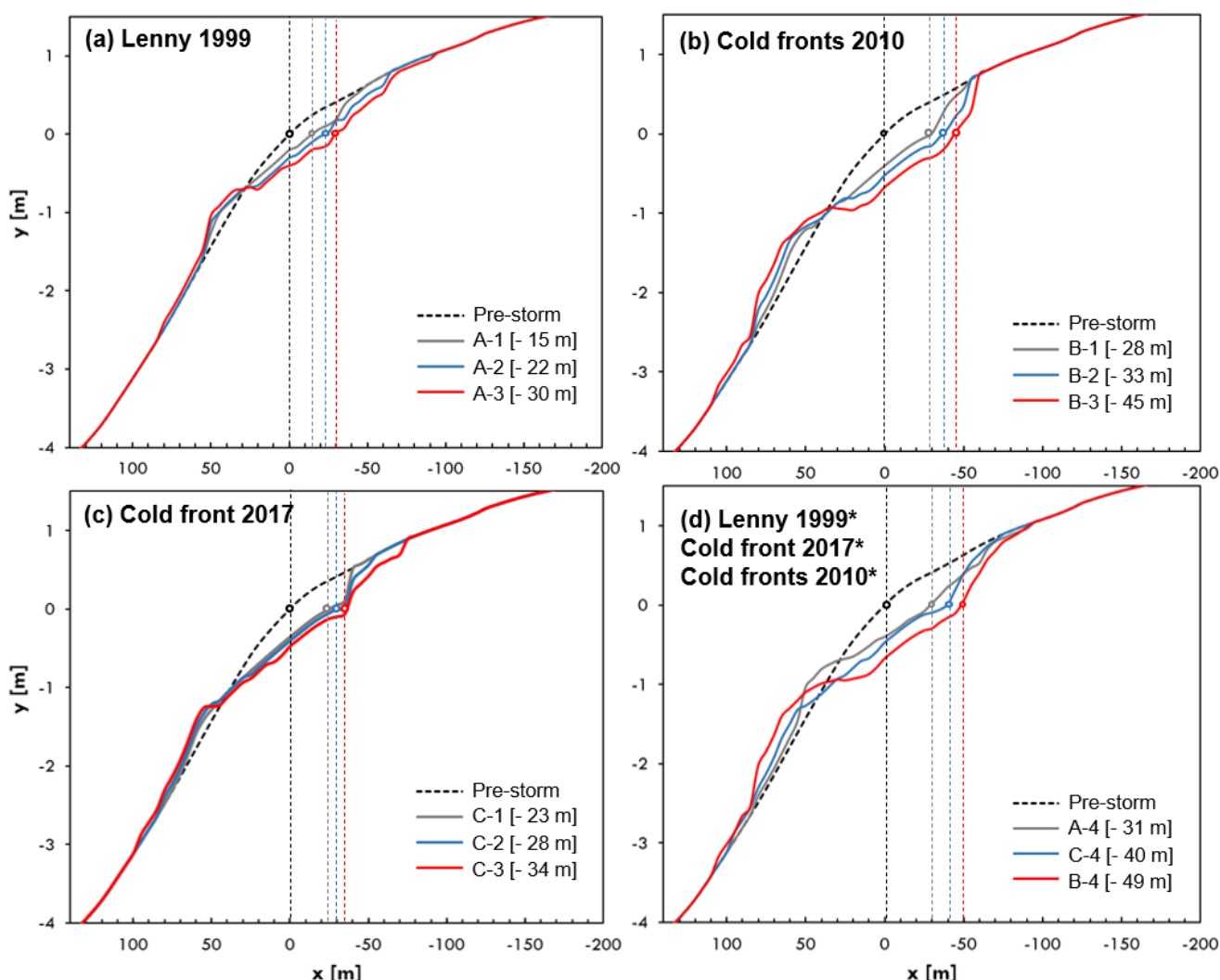

**Figure 7: Post-storm retreats with different SLR scenarios of Bocagrande's control profile for (a) Lenny 1999 (cases A-1, A-2, and A-3), (b) the cold fronts of 2010 (cases B-1, B-2, and B3), (c) the cold front of 2017 (cases C1, C2, and C3), and (d) the mentioned storms including high tide conditions \*(+0.25 m) (cases A4, B4, and C4).**

## 5 Discussion

We investigated the role of morphological changes on coastal flooding caused by storms and SLR on a microtidal dissipative beach. The numerical model was calibrated and validated with post-storm morphological data. This approach has not been widely used for studying the simultaneous impact of erosion and flooding in a coastal area with the previously described features. This is because among the reviewed and analysed investigations numerical models lacked a specific hydrodynamic and morphodynamic calibrations for the studied area (e.g., Annette et al., 2020). Following these ideas, the methods used in our study provide a better approximation of the actual flooding processes of a dissipative beach with microtidal regime such

as Bocagrande. Flood studies for this beach did not consider the effects of morphodynamics previously (Orejarena et al., 2019; Andrade et al., 2013). The morphodynamic and flooding processes cannot be decoupled because omitting the erosion leads to an underestimation of coastal flooding, as shown by the results of the present study and others (e.g., Alireza et al., 2020; Boyden et al., 2021).

The extreme episode caused by the sequence of cold fronts in 2010 represented the most critical condition of coastal erosion in the study period (Bernal et al., 2016). The succession of storms generated a progressive landward erosion owing to the slow recovery of the beach profile. The storms altered substantially the morphology of Bocagrande's control profile, resulting in a larger area of onshore erosion and a smaller area of accretion seawards (Figure 8). This cross-shore imbalance was caused by longitudinal sediment transport. In that sense, the post-storm profile was a consequence of the partial migration of sediments

from the beachfront toward the submerged beach, forming a longitudinal bar (also formed in different magnitudes after every evaluated scenario) that modified the hydrodynamic conditions of the study area. The presence of bars could play a fundamental role on dissipating energy from incoming waves and apparent friction. The formation (or migration) of these morphological features changes the location of the waves breaking point, and thus, new boundary conditions are established for the local morphodynamics. These results demonstrate the importance of using the approach of this research, since only

individual storms are evaluated typically for hydrodynamic and morphodynamic processes that involve highly energetic conditions. Because of the spatial resolution of the XBeach model (cell sizes of 1-3-5 m), bed forms below the order of one meter (e.g., ripples) are neglected. Seasonal changes in bed forms are also neglected since the time scale used for modelling is limited to few days at maximum.

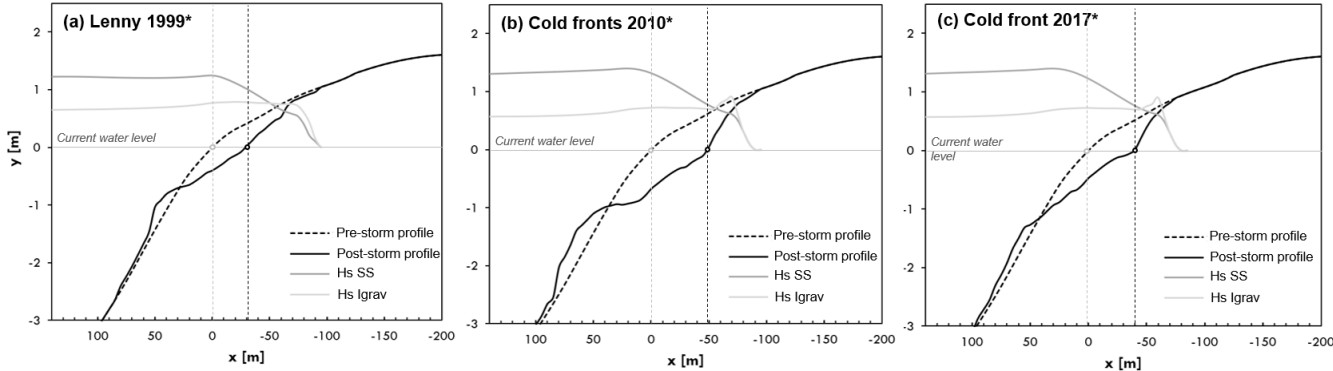


**Figure 8: Time-averaged SS and Infragravity (Igrav) waves and beach profiles before and after the storm for the most critical SLR condition (cases No. 4). Vertical dotted lines represent beachfront before and after the storm.**

Changes in eroded sections among the scenarios could be caused by two main factors: (i) storms duration and (ii) wave direction. In all simulated cases it was evident that the greatest impact on the coast (in terms of flooding and erosion) was

mainly attributable to the storm duration. Although Lenny 1999 ($H_{s\ max}$ = 2.76 m, two days) and the 2017 cold front ($H_{s\ max}$ =

3.53 m, five days) had the most energetic waves, the successive cold fronts of 2010 ($H_{s\ max\ averaged}$ = 1.87 m, 14 days) produced more erosion and flooding. Ortiz-Royero et al. (2013), Bernal et al. (2016) and Otero et al. (2016), who carried out studies evaluating the impact of storms on Colombian Caribbean beaches, also point towards this direction. They conclude that cold fronts (events which are commonly longer in time for the studied area) tend to cause more critical floods and beach loss. On

the other hand, slightly variations in wave direction between case studies could have a significant influence when waves are interacting with the multiple hard structures of Bocagrande. Reflection and refraction processes within the surf and swash zones are altered when the incident angle of incoming waves is modified, leading to changes in sediment transport patterns, and subsequently, the erosion/accretion outcome of the model. Small changes in wave directions acting for longer time windows could possibly cause differences in the final bathymetry of the beach for every case study. Moreover, the presence

of groins can induce alongshore sediment transport gradients, enhancing erosion at some locations. Given that the present study examined real conditions of storms, each of which had specific characteristics (i.e., $H_s$, $T_p$, $D_m$ and duration) that make it difficult to establish relationships between waves and flooding/erosion. We suggest an assessment of the influence of the various wave parameters on flooding and erosion by varying the water level and the duration of the storm for future studies.

The model results show that inside the surf zone the SS wave energy is dissipated while the infragravity wave energy grows due to nonlinear eergy transfer from high to low frequencies. Thus, the infragravity waves dominating nearshore hydrodynamics, consistent with the results of Conde et al. (2017b) (Figure 8). This result highlights the importance of involving infragravity waves in studies related to erosion and flooding of the coast, because those waves increase the oscillation of the run-up, causing further erosion (de Vries et al., 2007; Kamphuis, 1996). The erosion of dissipative beaches has been attributed

mainly to the forcing of the infragravity regime, owing to its predominance. Many authors highlight the importance of infragravity waves in sediment transport (e.g., Holman and Bowen, 1982; Carter et al., 1973) because they have been shown efficient in suspending sediments in surf and swash zones (Osborne and Rooker, 1999; Aagaard and Greenwood, 1994; Beach and Sternberg, 1991). Just there, suspended sediments are transported by currents, exacerbating coastal erosion. These effects are expected to intensify during extreme wave conditions because of increased infragravity wave energy (Senechal et al., 2011;

Ruggiero et al.; 2004; Ruessink, 1998). However, the results of Conde et al. (2017b) for this beach show a saturation of infragravity energy in the swash zone; therefore, this is also true for the run-up. It should be noted that the aforementioned studies did not consider hydrodynamic-morphodynamic interaction.

The present study also addresses the importance of sea level in predicting beach erosion and coastal flooding during storms.

Model results indicate greater flooding and retreat of the coastline as sea level increases. Historical storms evaluated with different SLR scenarios show the most critical situation for a greater SLR and the cold fronts sequence. An increase in sea level allows the energy dissipation of SS waves to occur more onshore, promoting coastline retreat. This reflects the strong vulnerability of the coast of Cartagena to erosion and flooding by potential threats such as (i) extreme waves generated by cold fronts and hurricanes, and (ii) a progressive increase in sea level, whose effects on the coasts strengthen when storms coincide

with high tides (Figure 9). Erosion and flooding processes during storms, combined with SLR, can cause critical alterations on a dissipative microtidal beach. This type of beach is common in many parts of the world (e.g., on the coasts of the Caribbean and Mediterranean seas and Gulf of Mexico). Previous studies have analysed the trends of the extreme waves at the Caribbean Sea based on wave reanalysis (Izaguirre et al., 2013; Reguero et al. 2013; Appendini et al., 2014). However, the study area does not show a clear positive trend and hence SLR seems to be a major threat for this site.


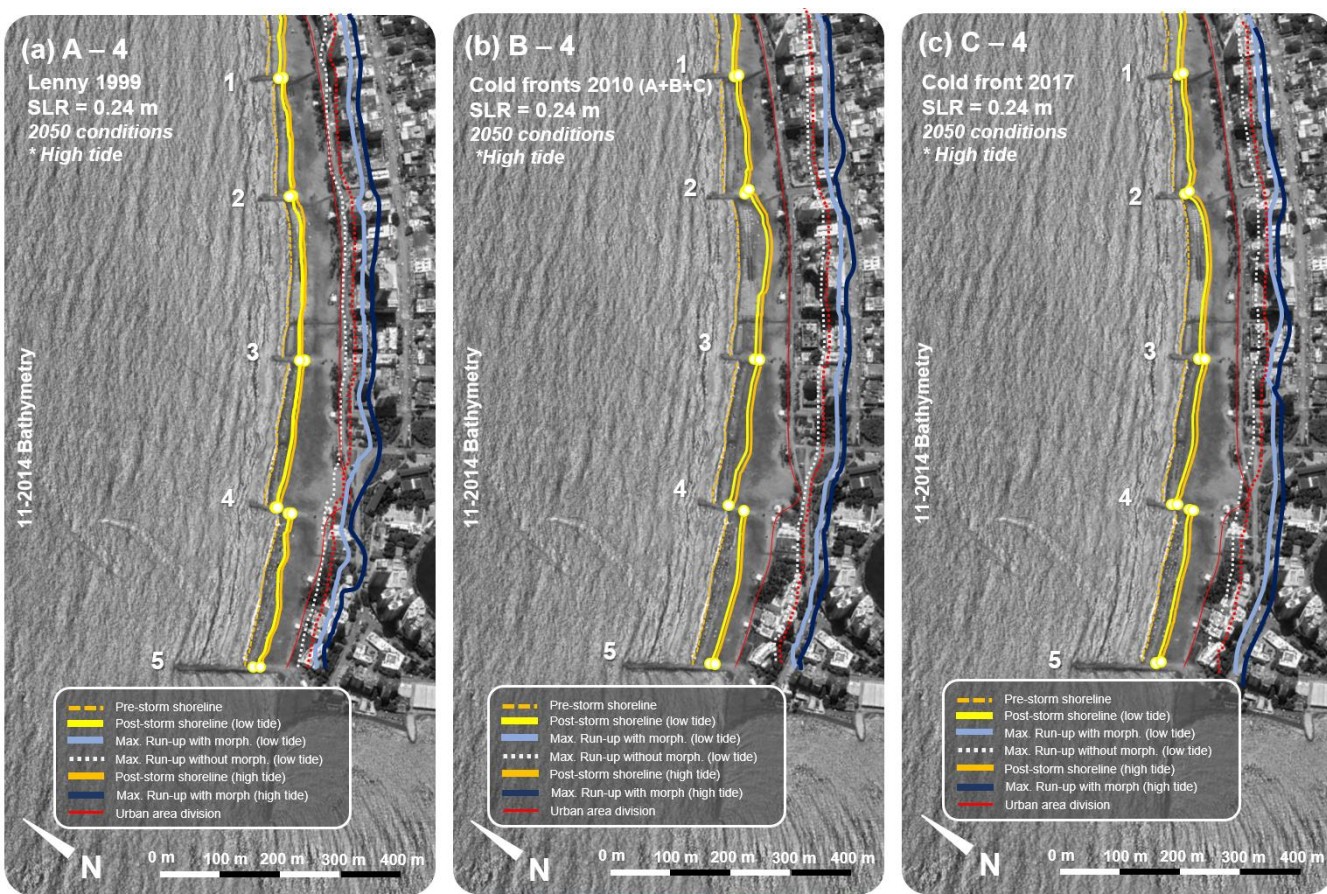

**Figure 9: Maximum flood reach and post-storm shorelines simulated by XBeach at low and high tide. Map base images retrieved from Google Earth © and modified by the authors.**

Moreover, the beach morphology in the study area was assumed to remain the same under the future scenarios, whereas the submerged and subaerial beach will be adjusted to changes caused by environmental factors and anthropogenic interventions outside the scope of this research. Recent studies suggest that the gradual SLR and extreme storms allow beaches to migrate and mitigate adverse effects (e.g., Cooper et al., 2020), but this approach should be carefully analysed since behind Bocagrande there is an urban area that would prevent a proper beach migration. (i.e., coastal squeeze). It is important to point that the

suitability of the present analysis may act on beaches further from equilibrium and hence the adverse effect may be underestimated. From the calibration process of XBeach, calculated bed level changes under storm and SLR conditions are a good estimation for the site. However, the numerical model does not consider the contribution of individual waves to morphodynamics when using the surfbeat mode. XBeach mainly uses the infragravity wave band to calculate morphodynamics, so the solving of diffraction processes (important in the presence of hard structures) can be limited.


The presented methodology could be also applied on beaches with different configurations, but some adjustments must be executed. In the case of reflective beaches, it is necessary to consider the non-hydrostatic mode of XBeach. Since short waves will be more relevant for the surf and swash zones dynamics on reflective beaches, the surfbeat mode could be limited for solving the hydrodynamic processes involved. For meso and macrotidal beaches, it is suggested to include the modelling of 430 different scenarios of astronomical tide. The modulation of the wave breaking point could contribute to an increase/decrease of the beach volume during storms.

## 6 Conclusions

Numerical models (SWAN and XBeach) were used to investigate the role of morphodynamics in predicting coastal flooding on a dissipative beach considering storm wave conditions and different sea level rise scenarios. The simulated cases, based in 435 wave records, corresponded to e Hurricane Lenny (1999), the cold front of December 2017, and the cold fronts sequence of March 2010. Moreover, the SLR projections for 2025 and 2050, and a special high-tide condition for the studied area were also included within the case studies.

The numerical results showed that flooding on microtidal dissipative beaches under storm scenarios should considered 440 morphodynamical modelling. Neglecting these processes could lead to an underestimation of coastal flooding by ~15%. A slow beach recovery after storms creates post-storm profiles with an erosion-accretion imbalance. Beach erosion can also increase due to alongshore sediment transport gradients induced by groins. Besides, erosion and flooding are intensified when SLR water levels are applied in the models. The most unfavourable condition was presented for extreme events that are contemporaneous with high tides. In this case, the increase in erosion and flooding is ~69% and ~65%, respectively, when 445 compared with the present conditions of sea level. Individual storms are typically evaluated to establish the threat to coastal zones, but the results of the present study suggest the need to evaluate scenarios considering storm sequence when the worst erosion and flooding scenario were found.

The present approach can be applied to other microtidal dissipative beaches in the world (e.g., on the coasts of the Caribbean 450 and Mediterranean seas and Gulf of Mexico). Accounting for beach morphodynamics for flooding prediction should be considered in early warning systems.

## 7 Acknowledgments

This study was partially funded by Minciencias (*Proyecto 121571250570 Convocatoria 712: "Turbulencia y mecanismos de disipación de energía de ondas gravitatorias e infragravitatorias en la zona de rompientes"*). The support of the Research
Office (DIDI) of Universidad del Norte is acknowledged.

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
