# Peer review of "The role of morphodynamics in predicting coastal flooding from storms on a dissipative beach with SLR conditions"

_Natural Hazards and Earth System Sciences, 2021_

## Referee Comment (RC1)

**The role of morphodynamics in predicting coastal flooding from storms on a dissipative microtidal beach with SLR conditions: Cartagena de Indias (Colombia).**

The authors present a study on the role of morphodynamic changes in the flooding of dissipative beaches with microtidal regime, using Cartagena de Indias as study case. Although the importance of the role of the role of morphodynamic changes has been stressed by other authors, the presented work is novel regarding the direct quantification of changes in the prediction of flooding for the conditions of the study site and under different scenarios of SLR.

Overall, the paper is well structured, with results being presented in a clear and organized manner. Some general comments regarding the inclusion of limitations of the adopted approach in the discussion section, and specific comments to increase overall comprehension of the work are given as follows.

General comments

[G1] The main parameter presented as calibration variable is *facua,* which is accompanied in Table 4 by the Chezy coefficient. However, the Chezy coefficient is fix, directly related to the sediment grain size, and the reader must assume that the value is supported by previous calibrations for the site. Thus, it is recommended to include a comment on how the value of the Chezy parameter relates to the actual characteristics of the beach, and, as it is presented as a morphodynamic parameter, it is also recommended to include a comment on how results might be affected by changes in this parameter.

[G2] The discussion section should be further completed commenting on the limitations and/or assumptions of the adopted approach. One example is the limitation of Xbeach to properly calculate morphodynamic processes related to short (individual) waves. The model is presented in such a way that the reader understands that it is solving completely both components, while the model mainly uses the infragravity wave band to calculate morphodynamics, while it calculates morphodynamics related to the short waves by adding the contribution of the short waves to the infragravity waves. This means that, for instance, that the model has some limitations solving the diffraction processes taking place in study areas where processes are affected by the presence groins. Some notice about this is given in lines 342-346 and 348-356.Another example is that the study does not consider the beach long term response (given enough accommodation space) to SLR. The study compares scenarios based on the assumption of a given reference morphology.

Specific comments

(line 48) See G2. It is recommended to specify that the morphological processes and run-up are calculated with the infragravity wave band accounting indirectly for the contribution of the short waves band.

(line 86) It is recommended to add also the usual surge range or magnitude of extreme surge, to give and idea about its relative contribution to the total water level at the beach.

(Table 1) To make information in the caption self-contained it is recommended to add the meaning of cases in italics.

(Line 159 / Table 2) what is the offshore depth of the XBeach domain? Please consider adding this information in the manuscript.

(Table 2 and related text) Is the resolution the same in alongshore and crosshore directions? It is recommended to specify it in the manuscript.

(line 176) See line 48. See G2. It is recommended to specify that the morphological and run-up processes are calculated with the infragravity wave band accounting indirectly for the contribution of the short waves band.

(line 182) Consider defining non-linear shallow water equations: NLSWE.

(table 5 and related text) In cases with multiple cold fronts, it is not clear whether the time between them is simulated or not. Also motivate in the text the intention behind scenarios A4, B4, and C4.

(figure 7) Specify what (*) stands for in the figure caption.

(line 336-337) Add reference of other authors pointing in the same direction (importance of duration to the magnitude of erosion and inundation).

---

## Referee Comment (RC2)

**MS No:** nhess-2021-210

**Full title**: The role of morphodynamics in predicting coastal flooding from storms on a dissipative microtidal beach with SLR conditions: Cartagena de Indias (Colombia)

**General remarks**

I read the m/s with great interest. Authors have investigated the extent of coastal flooding using previously calibrated two numerical models, SWAN and XBeach. Their main focus is to explore the flooding extend with and without morphodynamics in the simulations considering 5 scenarios (1 extreme event and 4 Cold front) together with sea level rise (SLR) and 3 scenarios with high tide. Approach and analyses support to derive their conclusions.

The content is interested for the NHESS readers. However, I found, the m/s needs careful improvements from abstract to conclusions. Therefore, I recommend moderate revision as suggested below before accepting for publication.

**Major comments**

1. Authors have simulated future scenarios representing 2025 and 2050 considering SLR only. It is not mentioned (ln 140 -145 or 2.3 Numerical Modelling), how SLR was implemented for the model water levels. Did you consider as water surface increase by SLR in both SWAN and XBeach models? Please state these clearly.
2. How reasonable to use the current condition of wave and wind for future scenarios of 2025 and 2050? This should be at least addressed in the discussion, we can not expect that wave and wind remain the same in future.
3. For identifying the events Lenny and Cold Front 2010 and 2017, authors have used wave time series from a virtual buoy based on the predicted WaveWatch3 NOAA data. It is not clear, which criterion was used to identify these events from the time series.
4. Under 2 Data and methods, it is more relevant to have 2.1 Study area and data and then 2.2 Approach or Methodology
5. Ln 133 when you mention 'Both events were selected…', it gives the impression that you have used only these two events to investigate the impacts of morphodynamics on coastal flooding, though you investigated in all selected scenarios. So remove this sentence and combine this paragraph with that of the below.
6. It is confusion using 'switch-off and -on sediment transport'. If you want to investigate erosion, you should activate bed level change: morphodynamics. Sediment transport itself does not mean erosion or accretion in your domain unless you have activated morphodynamics. Please correct this term.
7. Ln 235 remove description of Fig 7 from 3.1 to 3.4 and present results for all scenarios together at the end of this section: this would be very convenient for readers. For Fig 3-6, could you present one plot rather than three sub plots, putting all coastlines together (e.g., A-1, A-2 and A-3), and also indicate the initial coastline in each figure. The background could be in gray-scale and lines are in colours for a better visualization. Use 'present' instead of 'current conditions'.
8. At the end of the results section, I would recommend presenting one figure with results of all scenarios: x-axis, distance along the coast and y-axis: coastline position, that would give a better comparative impression of all scenarios than the values in Table 6.

9.  In table 6, please provide measured retreat for the present conditions (A1, B1 and C1). This will definitely give an added value for your modelling approach.
10. From the text, it is not clear, why does the maximum erosion section change depending on the scenarios? For example, in Lenny, the max erosion occurred in section 3-4 and 4-5 with waves Hs=2.76m, Dr=316, but similar waves with cold front (B1-A: Hs 2.66 and Dr 357.24) max erosion occurs in section 2-3. Could please discuss this difference in Discussion?

**Others comments**

1.  Please consider shortening the title: I find some words are not really necessary.
2.  The first two sentences in Abstract do not fit for an abstract, but for Introduction. The terminology, '..the simultaneous and individual effects of erosion and flooding scenarios…': this is not correct. You have investigated simultaneous effect of erosion and flooding, and then only flooding without erosion, but not erosion only without flooding. This term needs to corrected throughout the m/s.
3.  Ln 17 this study facilitates the construction of more precise models: do you want to develop precise models, or to accurate prediction of coastal flooding using numerical models?
4.  Ln 20 those numerical models were calibrated using field campaigns data: in this study, you used already calibrated models.
5.  Ln 20 'The results of this research indicate..' or you want to mention 'Results indicate..': short and sweet!
6.  Ln 22 Could you provide here quantitative value to indicate the increase of erosion and flooding by SLR.
7.  Ln 25 important or adverse
8.  Ln 28 I would write 'residence and industries', and what do you mean by 'exposed elements'
9.  Ln 40 How about increase in intensity and frequency of storm events as in recent IPCC report.
10. Ln 48 'This model solves…', this should go to the model description
11. Ln 51 Could you please use one terminology to indicate extreme events? You have used 'storms, extreme wave events, hurricane', is there a difference among these?
12. Ln 60 'serious erosion problems', are these related to storm events or chronical erosion?
13. Ln 63 after 50 meters include 'between January 2010 and January 2011'
14. Ln 67 no need 'it has been established that'
15. Ln 71 remove Per the above and I would write 'The main objective of the present work is..' Use 'morphodynamics' instead of 'morphodynamic changes', check throughout the m/s
16.  Ln 77 how it is important for the management of irrigation?
17. Ln 80 remove this sentence
18. Ln 85 Could you extend Fig 1d and show these six groynes
19. Ln 85 breaker zone instead of the area of breakers
20. Ln 87 mention neap and spring tidal ranges and the max tide occurred during your analysis period. Why did you select 0.24 m (section 3.4) high tide though it is 0.30 m (ln 87)?
21. Ln 88 could you explain, how did you construct the model bathymetries based on what data
22. Ln 97 readers would interest to find wave and wind characteristics within your analysis period: average conditions
23. Please indicate in Fig 1 caption, what are S1 to S5 locations
24. Ln 115 simply explain, how this database was adjusted, and provide a reference for the next sentence.

25. Ln 124 briefly explain how did you integrate here
26. Ln 112 mention the depth of your virtual buoy
27. Ln 113 reanalysis or model predicted series from NOAA
28. In Table 1 caption, remove star and indicate selected events in bold letters
29. In Table 2 caption, Y nodes instead of And nodes, also provide offshore depth of each model
30. Ln 175 XBeach was originally developed as a collaboration research among IHE Delft, US Army and Deltares, see end of Roelvink et al (2009), so not only Delft University of Technology
31. Ln 177 what is 'Igrav', pleas explain
32. Ln 182 what is NLSWE?
33. 187 check km$^2$
34. Ln 189 please provide depth range of each model
35. Ln 198 correct appearance of unit
36. In Table 4, where are the location of these profile from A to D, indicate them on Fig 1?
37. Ln 205 Do you want to switch off sediment transport or bed level update to avoid erosion/accretion
38. In Table 5, caption it should be 'evaluation of flooding..', this table is not easy to understand. Case study A1-A4, B1-B4.. and so on, are they related to profile name in previous table, otherwise please use a different notation. Under storm, what is meant by A+B+C? please present in a way that you do not have to repeat wave conditions, also in durations. Note, it is enough to use one decimal place for hs and tp and no decimals for direction.
39. No need the first sentence in Results: we know that you going to explain results here
40. In the approach section, you should clearly mention your scenarios, SLR and high tide study cases
41. Ln 238 please explain in Approach, how did you estimate the flooding extent based on your model results.
42. Please use same terminology to indicate morphodynamics, Ln 259: non-static bottom, ln 291:erosiove processes, ln 292 sed-on etc
43. Ln 287 '..that comprise the XBeach computational domain,…', does this mean, you can evaluate even beyond your computational domain?
44. Captions Fig 3-6, should be in unique way, as only the scenarios are changed. Use 'with and without morphodynamics' instead of sediment transport module on and off. We interest in processes, but not the modules of these models.
45. In Fig 7, I would be very interested to see 'post-storm' measured profile in each subplot. Then, you want to mention the caption as, 'Simulate and measured beach retreats along….'
46. Again please change everywhere, with and without morphodynamics rather sed-on and sed-off: we interest in processes not models.
47. In Table 6, where is the reference line for maximum penetration in urban area, please mention those things in approach or under different section of analysis parameters
48. Ln 315 this sentence is not clear, rephrase it (two time of caused by)
49. Ln 317 'as far as the authors….', nice to say: Our novelty approach..
50. Ln 322 'erosion' or morphodynamics during flooding
51. Fig 8 easy to understand with colour lines
52. Ln 351 'invigorating' or 'exacerbate'
53. Ln 378 where did you get 15%, I did not find this earlier in your results
54. Please write conclusion focussing: approach, main findings, relevance and applicability for other study areas
55. Please use standard reference format for NHESS, eg,

Aagaard, T. and Greenwood, B.: Suspended sediment transport and the role of infragravity waves in a barred surf zone, Marine Geology, 118(1–2), 23–48, https://doi.org/10.1016/0025-3227(94)90111-2, 1994.

Andrade, C. A., Thomas, Y. F., Lerma, A. N., Durand, P., and Anselme, B.: Coastal Flooding Hazard Related to Swell Events in Cartagena de Indias, Colombia. Journal of Coastal Research, 290(5), 1126–1136, 390 https://doi.org/10.2112/JCOASTRES-D-12-00028.1, 2013

Beach, R.A. and Sternberg, R.W.: Infragravity Driven Suspended Sediment Transport in the Swash, Inner and Outer-Surf Zone. Proceedings Coastal Sediments, 91, New York, ASCE, pp. 114-128, 1991.

---

## Author Comment (AC1)

**Response to Referee #1 comments:**

"The role of morphodynamics in predicting coastal flooding from storms on a dissipative microtidal beach with SLR conditions: Cartagena de Indias (Colombia)" (nhess-2021-210) by Jairo E. Cueto Fonseca et al.

**General Comments**

- **[GC1] The main parameter presented as calibration variable is facua, which is accompanied in Table 4 by the Chezy coefficient. However, the Chezy coefficient is fix, directly related to the sediment grain size, and the reader must assume that the value is supported by previous calibrations for the site. Thus, it is recommended to include a comment on how the value of the Chezy parameter relates to the actual characteristics of the beach, and, as it is presented as a morphodynamic parameter, it is also recommended to include a comment on how results might be affected by changes in this parameter.**

  RESPONSE: The value of the Chezy coefficient was previously determined by Cueto and Otero (2020) at this site. Such clarification will be incorporated on the revised manuscript as follows:

  "The Chezy friction coefficient is directly related to the sediments' characteristics of the studied beach, which is mostly constituted by fine sands with grain sizes within the range of 0.08 and 0.42 mm (Conde *et al.,* 2017a; b). According to the tests conducted by Cueto and Otero (2020), an increase (decrease) in the bottom friction through the Chezy coefficient would cause greater (lower) dissipation of the incident waves energy, leading to an underestimation (overestimation) of on Bocagrande's morphology fluctuations."

- **[GC2] The discussion section should be further completed commenting on the limitations and/or assumptions of the adopted approach. One example is the limitation of Xbeach to properly calculate morphodynamic processes related to short (individual) waves. The model is presented in such a way that the reader understands that it is solving completely both components, while the model mainly uses the infragravity wave band to calculate morphodynamics, while it calculates morphodynamics related to the short waves by adding the contribution of the short waves to the infragravity waves. This means that, for instance, that the model has some limitations solving the diffraction processes taking place in study areas where processes are affected by the presence groins. Some notice about this is given in lines 342-346 and 348-356.Another example is that the study does not consider the beach long term response (given enough accommodation space) to SLR. The study compares scenarios based on the assumption of a given reference morphology.**

  RESPONSE: We thank the reviewer for pointing out to the need of discussing the model limitations. We agree that the surf beat model does not incorporate intra-wave processes that are particularly important in the swash zone. On the other hand, the beach morphology in the study area was assumed to remain the same under the future scenario, whereas in reality the submerged and subaerial beach is adjusted to such changes. Recent studies suggest that the gradual SLR and extreme storms allow the beach to migrate and mitigate adverse effects (e.g., Cooper et al., 2020; Harley et al., 2021). The discussion will be extended towards this direction.

**Specific comments**

- **SC1. (line 48) See G2. It is recommended to specify that the morphological processes and run-up are calculated with the infragravity wave band accounting indirectly for the contribution of the short waves band.**

  RESPONSE: The revised manuscript highlights that both morphological processes and runup are calculate while assuming that the ingravity energy dominates in the swash zone. Stockdon et al. (2006) shows that dissipative beaches are dominated by infragravity energy owing to the saturation of the short waves band. The latter suggests that the surfbeat mode of XBeach is suitable for accounting the impact of storms under highly energetic wave conditions (as in this research). The non-hydrostatic mode can solve individual waves, but this was not used in the present work due to a significant increase in the computation cost.   Comment added but moved to the model description section (2.3.2).

- **SC2. (line 86) It is recommended to add also the usual surge range or magnitude of extreme surge, to give and idea about its relative contribution to the total water level at the beach.**

  RESPONSE: According to Andrade et al., 2013, the extreme surges are around the order of 0.2 m for the studied area. This information is now included in the manuscript.

- **SC3. (Table 1) To make information in the caption self-contained it is recommended to add the meaning of cases in italics.**

  RESPONSE: The selected cases are now in bold letters and the corresponding text has been included in Table 1 caption.

- **SC4. (Line 159 / Table 2) what is the offshore depth of the XBeach domain? Please consider adding this information in the manuscript.**

  RESPONSE: The offshore boundary for the XBeach and SWAN models are located at 6.5 m and 830 m, respectively. This information has been included in the revised manuscript (table and text).

- **SC5. (Table 2 and related text) Is the resolution the same in alongshore and crosshore directions? It is recommended to specify it in the manuscript.**

  RESPONSE: The resolution was specified in Table 2 for both models.

- **SC6. (line 176) See line 48. See G2. It is recommended to specify that the morphological and run-up processes are calculated with the infragravity wave band accounting indirectly for the contribution of the short waves band.**

  RESPONSE: See reply above (i.e., GC2)

- **SC7. (line 182) Consider defining non-linear shallow water equations: NLSWE.**

RESPONSE: The acronym NLSWE is now defined in the manuscript.

- **SC8. (table 5 and related text) In cases with multiple cold fronts, it is not clear whether the time between them is simulated or not. Also motivate in the text the intention behind scenarios A4, B4, and C4.**

RESPONSE: The time between the cold fronts A, B and C of 2010 was simulated using the stationary mode of XBeach as wave conditions during those intervals were low-energetic (~0.7 – 0.8 m). This information has been now included in the text. The intention behind scenarios A4, B4 and C4 was explained.

- **SC9. (figure 7) Specify what (*) stands for in the figure caption.**

RESPONSE: The * stands for high tide conditions (+0.25 m) in each case. The figure's caption has been revised accordingly.

- **SC10. (line 336-337) Add reference of other authors pointing in the same direction (importance of duration to the magnitude of erosion and inundation).**

RESPONSE: Additional references (e.g., Ortiz-Royero et al. (2013), Bernal et al. (2016) and Otero et al. (2016)) have been included.

---

## Author Comment (AC2)

**Response to Referee #2 comments:**

"The role of morphodynamics in predicting coastal flooding from storms on a dissipative microtidal beach with SLR conditions: Cartagena de Indias (Colombia)" (nhess-2021-210) by Jairo E. Cueto Fonseca et al.

**General Comments [GC]**

1. **Authors have simulated future scenarios representing 2025 and 2050 considering SLR only. It is not mentioned (ln 140 -145 or 2.3 Numerical Modelling), how SLR was implemented for the model water levels. Did you consider as water surface increase by SLR in both SWAN and XBeach models? Please state these clearly**

RESPONSE: The sea levels of 2025 and 2050 for Cartagena, based on the work by Orejarena et al. (2019), were implemented in both SWAN and XBeach models. The SLR was added to the current water depth for the different cases. The implementation of SLR is now described in Section 3.2.2 (morphodynamics and flooding).

2. **How reasonable to use the current condition of wave and wind for future scenarios of 2025 and 2050? This should be at least addressed in the discussion, we can not expect that wave and wind remain the same in future.**

RESPONSE: Previous studies have analyzed the trends of the extreme waves at the Caribbean Sea based on wave reanalysis (Izaguirre et al., 2013; Reguero et al. 2013; Appendini et al., 2014). However, the study area does not show a clear positive trend and hence SLR seems to be a major threat for this site. This comment will be added and extended within the discussion section.

3. **For identifying the events Lenny and Cold Front 2010 and 2017, authors have used wave time series from a virtual buoy based on the predicted WaveWatch3 NOAA data. It is not clear, which criterion was used to identify these events from the time series.**

RESPONSE: The events were identified from contrasting the official dates with the wave series extracted from the virtual buoy VB01. Official dates were taken from the reports of the Center of Oceanographic and Hydrographic Research of Colombia (CIOH) and the following authors: Ortiz-Royero, 2012; Ortiz-Royero et al., 2013; Bernal et al., 2016 and Otero et al., 2016. These authors also pointed that the listed extreme events caused the biggest negative impact over the Caribbean coasts during the last four decades. This is cleared in the manuscript.

4. **Under 2 Data and methods, it is more relevant to have 2.1 Study area and data and then 2.2 Approach or Methodology**

RESPONSE: Following the reviewer's suggestion, the manuscript structure has been revised. Description of study area is now a complete section (2) and methodology is now section 3 (includes 3.1 Selected events and 3.2 Numerical modelling - 3.2.1 From deep waters to the coast: Wave propagation using SWAN and 3.2.2 Morphodynamics and flooding: XBeach).

5. **Ln 133 when you mention 'Both events were selected…', it gives the impression that you have used only these two events to investigate the impacts of morphodynamics on coastal flooding, though you investigated in all selected scenarios. So remove this sentence and combine this paragraph with that of the below.**

RESPONSE: We thank the reviewer for this suggestion. The sentence was deleted and the indicated paragraphs were combined for a better understanding of the section.

**6. It is confusion using 'switch-off and -on sediment transport'. If you want to investigate erosion, you should activate bed level change: morphodynamics. Sediment transport itself does not mean erosion or accretion in your domain unless you have activated morphodynamics. Please correct this term.**

RESPONSE: The reviewer is right. We were referring to the morphological updating (bed level change) of XBeach. The term was corrected in the whole manuscript (also stated in the specific comments) as suggested.

**7. Ln 235 remove description of Fig 7 from 3.1 to 3.4 and present results for all scenarios together at the end of this section: this would be very convenient for readers. For Fig 3-6, could you present one plot rather than three sub plots, putting all coastlines together (e.g., A1, A-2 and A-3), and also indicate the initial coastline in each figure. The background could be in gray-scale and lines are in colours for a better visualization. Use 'present' instead of 'current conditions'.**

RESPONSE: The description of Figure 7 will be removed from the sections 3.1 to 3.4. We will follow the suggestion of the reviewer of having a final section (3.5) that comprehends the description of the control profile behavior under all scenarios. Initial coastline will be included and also the color scheme will be changed to gray scale for a better visualization. On the other hand, we think that having one plot per storm rather than three subplots could cause visual stress to the audience. That single plot would have at least ten lines (including the initial coastline); in that way coastline migration and flooding (with and without morphodynamics) results could not be well differentiated by readers. We will keep the three subplots per storm that show each case separately.

**8. At the end of the results section, I would recommend presenting one figure with results of all scenarios: x-axis, distance along the coast and y-axis: coastline position, that would give a better comparative impression of all scenarios than the values in Table 6.**

RESPONSE: This recommendation is welcomed, thanks to the reviewer. The Figure is now included within the manuscript for a better comparative impression of all scenarios; it is a good complement to Table 6.

**9. In table 6, please provide measured retreat for the present conditions (A1, B1 and C1). This will definitely give an added value for your modelling approach.**

RESPONSE: The measured retreat for the present conditions is now included in Table 6 as suggested.

**10. From the text, it is not clear, why does the maximum erosion section change depending on the scenarios? For example, in Lenny, the max erosion occurred in section 3-4 and 4-5 with waves Hs=2.76m, Dr=316, but similar waves with cold front (B1-A: Hs 2.66 and Dr 357.24) max erosion occurs in section 2-3. Could please discuss this difference in Discussion?**

RESPONSE: Changes in eroded sections among the scenarios could be caused by two main factors: i) wave direction and ii) storms duration. These slightly variations in wave direction between the reviewer's commented scenarios could have a significant influence when waves are interacting with the multiple hard structures of Bocagrande. Reflection and refraction processes (also currents) within the surf and swash zones are altered when the incident angle of incoming waves is modified, leading to changes in sediment transport patterns, and subsequently, the erosion/accretion outcome of the model. On the other hand, storm duration is different for Lenny, the successive cold fronts of 2010 and the cold front of 2017. Small changes in wave directions acting for longer time windows could possibly cause differences in the final bathymetry of the beach. This will be addressed in the discussion section as suggested.

**Specific Comments [SC]**

**1. Please consider shortening the title: I find some words are not really necessary.**
RESPONSE:The title was shortened: "The role of morphodynamics in predicting coastal flooding from storms on a dissipative beach with SLR conditions".

**2. The first two sentences in Abstract do not fit for an abstract, but for Introduction. The terminology, '..the simultaneous and individual effects of erosion and flooding scenarios...': this is not correct. You have investigated simultaneous effect of erosion and flooding, and then only flooding without erosion, but not erosion only without flooding. This term needs to corrected throughout the m/s.**
RESPONSE: Following the reviewer's suggestions we have revised the abstract as follows: "We investigate the role of morphodynamic changes in the flooding of a dissipative beach with microtidal regime for both current and sea level rise scenarios. By considering beach morphodyanmics and flood processes associated with highly energetic waves, the study allows to evaluate threats to coastal zones. Coupling of SWAN and XBeach models are employed to propagate offshore wave conditions to the swash zone, estimating morphological changes and flooding associated to wave conditions during cold fronts and hurricanes that affected Cartagena de Indias. Numerical models were calibrated from a previous research in the studied area. The results indicate that flooding on microtidal dissipative beaches under extreme wave conditions should be approached by considering beach morphodynamics, because ignoring them can underestimate flooding by ~15%. Moreover, beach erosion and flooding are intensified by sea level rise, resulting in the most unfavorable condition when extreme events are contemporaneous with high tides. In this case, the increase in erosion and flooding is ~69% and ~65%, respectively, when compared with the present conditions of sea level."

**3. Ln 17 this study facilitates the construction of more precise models: do you want to develop precise models, or to accurate prediction of coastal flooding using numerical models?**
RESPONSE: We have revised the text to clarify that we want to improve the model predictivity and not develop new models.

**4. Ln 20 those numerical models were calibrated using field campaigns data: in this study, you used already calibrated models.**
RESPONSE: The reviewer appreciation is right; this is now clarified. SWAN and XBeach were previously calibrated by Cueto and Otero (2020) for the studied area. In further sections of the manuscript this is also specified.

**5. Ln 20 'The results of this research indicate..' or you want to mention 'Results indicate..': short and sweet!**
RESPONSE: Fixed.

**6. Ln 22 Could you provide here quantitative value to indicate the increase of erosion and flooding by SLR.**

RESPONSE: "Moreover, beach erosion and flooding are intensified by sea level rise, resulting in the most unfavorable condition when extreme events are contemporaneous with high tides. In this case, the increase in erosion and flooding is ~69% and ~65%, respectively, when compared with the present conditions of sea level." Added.

**7. Ln 25 important or adverse**
RESPONSE: Clarified: Adverse.

**8. Ln 28 I would write 'residence and industries', and what do you mean by 'exposed elements'**
RESPONSE: The sentence was simplified as suggested: "In highly urbanized coastal areas, such as Cartagena de Indias (Colombia), where residences and industries are located near the coast, such storms generally damage or destroy the infrastructure." The expression "exposed elements" referred to the infrastructure near the coast; it was changed.

**9. Ln 40 How about increase in intensity and frequency of storm events as in recent IPCC report.**
RESPONSE: Thanks to the reviewer for the suggestion. A new comment was included and referenced within the manuscript as it follows "Among the most important factors for the prediction are: (i) increase in sea level associated with climate change, which increases the exposure of the coast during extreme events; (ii) increase in intensity and frequency of storms events (IPCC, 2021); (iii) beach erosion; (iv) flooding during and after a storm (Elsayed and Oumeraci, 2016)."

**10. Ln 48 'This model solves…', this should go to the model description.**
RESPONSE: The sentences will be moved to the model description section as requested.

**11. Ln 51 Could you please use one terminology to indicate extreme events? You have used 'storms, extreme wave events, hurricane', is there a difference among these?**
RESPONSE: In line 51 we were particularly addressing extreme waves. The term "extreme wave events" is replaced by "extreme waves". Following your suggestion, we changed the other mentioned terms by "storms" in the whole article.

**12. Ln 60 'serious erosion problems', are these related to storm events or chronical erosion?**
RESPONSE: Those serious erosion problems are chronical along the Colombian Caribbean coast but specially augmented by storm impacts as stated by Rangel-Buitrago et al. (2015) and Otero et al. (2016). This information will be incorporated in the revised manuscript.

**13. Ln 63 after 50 meters include 'between January 2010 and January 2011'**
RESPONSE: Included.

**14. Ln 67 no need 'it has been established that'**
RESPONSE: The text has been removed.

**15. Ln 71 remove Per the above and I would write 'The main objective of the present work is..' Use 'morphodynamics' instead of 'morphodynamic changes', check throughout the m/s**
RESPONSE: Done.

**16. Ln 77 how it is important for the management of irrigation?**
RESPONSE: The word irrigation was a translation error and hence will be removed from the revised manuscript.

**17. Ln 80 remove this sentence**
RESPONSE: The sentence was removed as a new structure was made (general comment 4).

**18. Ln 85 Could you extend Fig 1d and show these six groynes**
RESPONSE: The figure was extended as suggested and now shows the six groins within the Bocagrande studied area.

**19. Ln 85 breaker zone instead of the area of breakers**
RESPONSE: Changed.

**20. Ln 87 mention neap and spring tidal ranges and the max tide occurred during your analysis period. Why did you select 0.24 m (section 3.4) high tide though it is 0.30 m (ln 87)?**
RESPONSE: The selected high-tide was 0.25 m, not 0.24 m (which is the SLR in this case). Anyways, we selected this value since 0.25 m is the most typical high-tide magnitude for Bocagrande and we wanted to evaluate this specific scenario.

**21. Ln 88 could you explain, how did you construct the model bathymetries based on what data**
RESPONSE: Detailed bathymetries were measured along with control profiles in field campaigns developed before and after a succession of cold fronts between November 2014 and February 2015. These bathymetries were constructed using data from an ODOM Hydrotrac 2 single beam echosounder (submerged areas) and a high-resolution LIDAR topography (dry beach and urban areas). All vertical heights in the Digital Terrain Model (DTM) were referenced to average low tides of syzygy (MLWS) and horizontal coordinates were referenced to UTM 18N, following the standards of the International Hydrographic Organization (IHO). This explanation was included in the manuscript as suggested.

**22. Ln 97 readers would interest to find wave and wind characteristics within your analysis period: average conditions**
RESPONSE: Detailed wave characteristics were explained for the analysis period to the readers. The averaged significant Sea-Swell wave heights during the wet season field campaign at S1, S2, S3, S4 and S5 were 1.2, 0.4, 0.4, 0.3 and 0.2 m, respectively; for the dry season campaign these values were 1.7, 0.7, 0.6, 0.5 and 0.4 m, respectively. Peak periods averaged 8 s (wet season campaign) and 9 s (dry season campaign) at the outermost sensor (S1). The most incident wave direction was north-northeast for both measuring periods. General wind remarks for the studied

area are already mentioned in the manuscript, but specific measurements of wind were not executed during the field campaigns.

**23. Please indicate in Fig 1 caption, what are S1 to S5 locations**
RESPONSE: The nomenclature is now explained in the caption, where S1 is a pressure sensor (RBR); S2 to S5 are current meters (Aquadopp). Also, the readers are referred to Cueto and Otero (2020) to see a detailed description of location, depth and measurement rates for the sensors.

**24. Ln 115 simply explain, how this database was adjusted, and provide a reference for the next sentence.**
RESPONSE: The approach is briefly described and the reference for the next sentence was included as it follows: "Vega (2017) adjusted these wave series were to the conditions of the Colombian Caribbean by using a hybrid methodology of wave reanalysis scale refinement. This approach included the effect of specific spectral wave, wind and bottom roughness parameters for the studied area after conducting a sensitivity analysis. It is well known that the wave reanalysis information tends to underestimate wave heights associated with extreme events within the Caribbean (Ortiz-Royero, 2009). However, with the adjustment of Vega (2017), errors do not exceed 5% for significant wave height calculations."

**25. Ln 124 briefly explain how did you integrate here**
RESPONSE: We apologize, the word "integrate" was a mistake from the Spanish to English translation. The last sentences of the paragraph were changed to have a better explanation of the idea. "To investigate the combined effects of erosion and flooding following the impact of storms in future scenarios with higher sea levels, the sea level rise (SLR) conditions for the coming decades at Bocagrande predicted and discussed by Orejarena et a. (2019) will be added when selected events are modelled."

**26. Ln 112 mention the depth of your virtual buoy**
RESPONSE: The depth of the virtual buoy was included (800 m).

**27. Ln 113 reanalysis or model predicted series from NOAA**
RESPONSE: Clarified: model predicted series from NOAA.

**28. In Table 1 caption, remove star and indicate selected events in bold letters**
RESPONSE: Suggestion accepted. Selected events are in bold letters and the star was removed.

**29. In Table 2 caption, Y nodes instead of And nodes, also provide offshore depth of each model**
RESPONSE: The typo mistake was corrected. Y [nodes] is now displayed in Table 2. The offshore domain of each model was also provided.

**30. Ln 175 XBeach was originally developed as a collaboration research among IHE Delft, US Army and Deltares, see end of Roelvink et al (2009), so not only Delft University of Technology**

RESPONSE: The mention of Delft University of Technology was erased. If we mention this University we have to make a big list of institutions (IHEDelft, Deltares, US Army Corp of Engineers, University of Plymouth, University of Western Australia, etc.) having a denser paragraph. We think that the reference of the original paper of the model by itself (Roelvink et al., 2009) is better in this specific case (it covers all government and non-government institutions).

**31. Ln 177 what is 'Igrav', pleas explain**
RESPONSE: Igrav stands for infragravity wave. The term was replaced.

**32. Ln 182 what is NLSWE?**
RESPONSE: NLSWE stands for non-linear shallow water equations. The term is now explained in the text.

**33. 187 check km2**
RESPONSE: Format changed as suggested.

**34. Ln 189 please provide depth range of each model**
RESPONSE: The depth range of each model was provided for the reader in Table 2.

**35. Ln 198 correct appearance of unit**
RESPONSE: The appearance of the unit was changed as suggested.

**36. In Table 4, where are the location of these profile from A to D, indicate them on Fig 1?**
RESPONSE: The location of the profiles was indicated in Figure 1 and Figure 2 as "control profile".

**37. Ln 205 Do you want to switch off sediment transport or bed level update to avoid erosion/accretion**
RESPONSE: The bed level update was turned off (morphological updating). Now this is clarified in the text.

**38. In Table 5, caption it should be 'evaluation of flooding..', this table is not easy to understand. Case study A1-A4, B1-B4.. and so on, are they related to profile name in previous table, otherwise please use a different notation. Under storm, what is meant by A+B+C? please present in a way that you do not have to repeat wave conditions, also in durations. Note, it is enough to use one decimal place for hs and tp and no decimals for direction.**
RESPONSE: The caption of Table 5 was revised as suggested and the whole table has a new structure that is easier to understand for the readers. Case studies are not related to the nomenclature shown in Table 4, so the latter was modified for not causing confusion. "A+B+C" was also replaced by X, Y and Z, which represent the three sucessive cold fronts of 2010. This nomenclature now is also explained in section 3.1 for a clearer interpretation. Repeated wave conditions and durations were deleted. Decimal place suggestion was accepted and changed within the table.

**39. No need the first sentence in Results: we know that you going to explain results here**
RESPONSE: The first sentence was deleted.

**40. In the approach section, you should clearly mention your scenarios, SLR and high tide study cases**
RESPONSE: This text is now included as suggested: "For the hydro-morphodynamic modelling with XBeach, the case studies shown in Table 5 were established. Case studies are derived from the selected events: Lenny 1999 ("A" cases), the cold fronts of 2010 ("B" cases – that include the cold fronts X, Y and Z) and the cold front of 2017 ("C" cases). Present conditions of sea level (cases A1, B1 and C1) and the future projections of SLR for 2025 (+0.11 m – cases A2, B2 and C2) and 2050 (+0.24 m – cases A3, B3 and C3), addressed by Orejarena et al. (2019), were included. Scenarios A4, B4 and C4 were set to analyse the effect of high tides on erosion and flooding processes combined with SLR. The morphological updating in XBeach was turned on and off for each case study. In this way, the influence of erosive processes on flooding was checked. The models included a non-erodible layer to simulate the hard structures present on Bocagrande beach. The time between the cold fronts X, Y and Z of 2010 was simulated using the stationary mode of XBeach as wave conditions during those intervals were low-energetic (~0.7 – 0.8 m). The sea levels of 2025 and 2050 for Cartagena were allocated at the offshore boundary as water level forcing. The SLR was added to the water level input for the different case studies conditions. This approach of water levels was also used when setting up the SWAN model for the studied area."

**41. Ln 238 please explain in Approach, how did you estimate the flooding extent based on your model results.**
RESPONSE: Flooding extent in all scenarios is estimated from the run-up output that XBeach calculates directly. This will be clarified when describing the methodology (3.2.2 Morphodynamics and flooding: XBeach).

**42. Please use same terminology to indicate morphodynamics, Ln 259: non-static bottom, ln 291:erosiove processes, ln 292 sed-on etc**
RESPONSE: The same terminology is now used as suggested.

**43. Ln 287 '..that comprise the XBeach computational domain,…', does this mean, you can evaluate even beyond your computational domain?**
RESPONSE: The sentence was erased to avoid confusion.

**44. Captions Fig 3-6, should be in unique way, as only the scenarios are changed. Use 'with and without morphodynamics' instead of sediment transport module on and off. We interest in processes, but not the modules of these models.**
RESPONSE: The captions were changed accordingly.

**45. In Fig 7, I would be very interested to see 'post-storm' measured profile in each subplot. Then, you want to mention the caption as, 'Simulate and measured beach retreats along….'**
RESPONSE: Thanks to the reviewer for the suggestion, but sadly there is no information of measured post-storm profiles after the selected scenarios. Beach profiles were measured only during the field campaigns of 2014 and 2015.

**46. Again please change everywhere, with and without morphodynamics rather sed-on and sedoff: we interest in processes not models.**
RESPONSE: Changed in the whole manuscript.

**47. In Table 6, where is the reference line for maximum penetration in urban area, please mention those things in approach or under different section of analysis parameters**
RESPONSE: Thanks to the reviewer for the suggestion. This reference line will be highlighted in the figures and mentioned in approach for a better interpretation of Table 6.

**48. Ln 315 this sentence is not clear, rephrase it (two time of caused by)**
RESPONSE: The sentence was clarified for the readers: "The results of the Xbeach model show the morphological changes and flooding caused by the effect of storms and SLR on a microtidal dissipative beach".

**49. Ln 317 'as far as the authors….', nice to say: Our novelty approach.**
RESPONSE: Changed.

**50. Ln 322 'erosion' or morphodynamics during flooding**
RESPONSE: Morphodynamics is the right term, changed.

**51. Fig 8 easy to understand with colour lines**
RESPONSE: We consider that the black and gray color contrast does not cause any confusion or visual stress to the reader in this particular figure. It is easy to distinguish between the SS and IG waves, and the pre and post-storm profiles. The figure remains with the original color scheme.

**52. Ln 351 'invigorating' or 'exacerbate'**
RESPONSE: Term corrected. It was "exacerbating".

**53. Ln 378 where did you get 15%, I did not find this earlier in your results**
RESPONSE: The 15% was obtained by comparing the averaged differences between the maximum extension points of the run-up with and without morphodynamis. This will be clarified in the manuscript.

**54. Please write conclusion focussing: approach, main findings, relevance and applicability for other study areas**
RESPONSE: Thanks for the comment. The conclusion section will be reorganized with the suggested structure.

**55. Please use standard reference format for NHESS**
RESPONSE: Reference format has been changed following NHESS guideline.

---

## Author Response (AR2)

**Response to Editor comments:**

"The role of morphodynamics in predicting coastal flooding from storms on a dissipative microtidal beach with SLR conditions: Cartagena de Indias (Colombia)" (nhess-2021-210) by Jairo E. Cueto Fonseca et al.

**General Comments**

**a) The role of morphodynamics in terms of bed forms on the resulting frictions**

RESPONSE: Because of the spatial resolution of the model (cell sizes of 1-3-5 m), bed forms below the order of one meter (e.g., ripples) are neglected. Seasonal changes in bed forms are also neglected since the time scale used for modelling is limited to few days at maximum. However, the cross-sections exposed in Figures 7 and 8 showed the formation of a longitudinal bar after every evaluated scenario. This morphological feature is generated by the offshore-directed sediment transport from the beachfront to the submerged beach. The presence of bars could play a fundamental role on dissipating energy from incoming waves and apparent friction. The formation (or migration) of bars modifies the hydrodynamics of Bocagrande, changing the location of the breaking point, and thus, establishing new boundary conditions for the local morphodynamics.

These comments were added in the discussion section (second paragraph: lines 350 to 363).

**b) The suitability of the analysis for combining projections of sea level rise and wave conditions, particularly considering that future wave conditions compounded by a different mean sea level may act on beaches further from equilibrium and therefore the model suitability should be also considered.**

RESPONSE: The suitability of the analysis and modelling strategy using XBeach when combining sea level rise and wave projections is now treated in the last paragraph (expanded) of the discussion section as it follows:

*" Moreover, the beach morphology in the study area was assumed to remain the same under the future scenarios, whereas the submerged and subaerial beach will be adjusted to changes caused by environmental factors and anthropogenic interventions outside the scope of this research. Recent studies suggest that the gradual SLR and extreme storms allow beaches to migrate and mitigate adverse effects (e.g., Cooper et al., 2020), but this approach should be carefully analysed since behind Bocagrande there is an urban area that would prevent a proper beach migration. (i.e., coastal squeeze). It is important to point that the suitability of the present analysis may act on beaches further from equilibrium and hence the adverse effect may be underestimated. From the calibration process of XBeach, calculated bed level changes under storm and SLR conditions are a good estimation for the site. However, the numerical model does not consider the contribution of individual waves to morphodynamics when using the surfbeat mode. XBeach mainly uses the infragravity wave band to calculate morphodynamics, so the solving of diffraction processes (important in the presence of hard structures) can be limited."*

*c) Potential for extrapolating the approach to other beach types such as reflective or macro tidal."*

RESPONSE: The presented methodology could be also applied on beaches with different configurations, but some adjustments must be executed. In the case of reflective beaches, it is necessary to consider the non-hydrostatic mode of XBeach. Since short waves will be more relevant for the surf and swash zones dynamics on reflective beaches, the surfbeat mode could be limited for solving the hydrodynamic processes involved. For meso and macrotidal beaches, it is suggested to include the modelling of different scenarios of astronomical tide. The modulation of the wave breaking point could contribute to an increase/decrease of the beach volume during storms.

These comments were added at the end of the discussion section.